# SCALING PROBABILISTIC CIRCUITS VIA DATA PARTITIONING

## ABSTRACT

Probabilistic circuits (PCs) enable us to learn joint distributions over a set of random variables and to perform various probabilistic queries in a tractable fashion. Though the tractability property allows PCs to scale beyond non-tractable models such as Bayesian Networks, scaling training and inference of PCs to larger, real-world datasets remains challenging. To remedy the situation, we show how PCs can be learned across multiple machines by recursively partitioning a distributed dataset, thereby unveiling a deep connection between PCs and federated learning (FL). This leads to federated circuits (FCs)—a novel and flexible federated learning (FL) framework that (1) allows one to scale PCs on distributed learning environments (2) train PCs faster and (3) unifies for the first time horizontal, vertical, and hybrid FL in one framework by re-framing FL as a density estimation problem over distributed datasets. We demonstrate FC's capability to scale PCs on various large-scale datasets. Also, we show FC's versatility in handling horizontal, vertical, and hybrid FL within a unified framework on multiple classification tasks.

## 1 INTRODUCTION

Probabilistic Circuits (PCs) are a family of models that provide tractable inference for various probabilistic queries (Poon & Domingos, 2011; Choi et al., 2020). This is achieved by representing a joint distribution by a computation graph on which certain structural properties are imposed. While PCs offer significant computational advantages over traditional probabilistic models such as Bayesian networks (Pearl, 1985), further performance gains can be realized by optimizing the compactness of PC representations and tailoring them to specific hardware architectures (Peharz et al., 2020a; Liu et al., 2024). However, another natural way to scale up PCs by distributing the model over multiple machines is so far underexplored. While models like neural networks can be partitioned over multiple machines with relatively low efforts, partitioning PCs is more challenging as they come with certain structural constraints to ensure the validity of the represented joint distribution. Interestingly, we find an inherent connection between the structure of PCs and the paradigm of federated learning (FL). In PCs, sum nodes combine probability distributions over the same set of variables via a mixture. This resembles the horizontal FL (Konečnỳ et al., 2016; Li et al., 2020) setting, where all clients hold the same features but different samples. In contrast, the case of vertical FL (Yang et al., 2019; Wu et al., 2020) in which the same samples are shared, but features are split across clients, can be linked to the product nodes used in PCs, which combine distributions of a disjoint set of variables. Consequently, the hybrid FL (Zhang et al., 2020) setting, where both samples and features are separated across clients, can be represented by a combination of sum and product nodes. Thus, PCs are well positioned to connect all three FL settings in a unified way – an endeavor considered hard to achieve in the FL community (Li et al., 2023a; Wen et al., 2023).

As a result of this connection, we introduce *federated circuits (FCs)*, a novel FL framework that re-frames FL as a density estimation problem over a set of datasets distributed over multiple machines (subsequently called clients). FCs naturally handle all three FL settings and, therefore, provide a flexible way of scaling up PCs by learning a joint distribution over a dataset arbitrarily partitioned across a set of clients (see Fig. 1 for an illustration). Imposing the same structural properties as for PCs, FCs achieve tractable computation of probabilistic queries like marginalization and conditioning across multiple machines. To this end, we propose a highly communication-efficient learning algorithm that leverages the semi-ring structure within the design of FCs. Our experimental

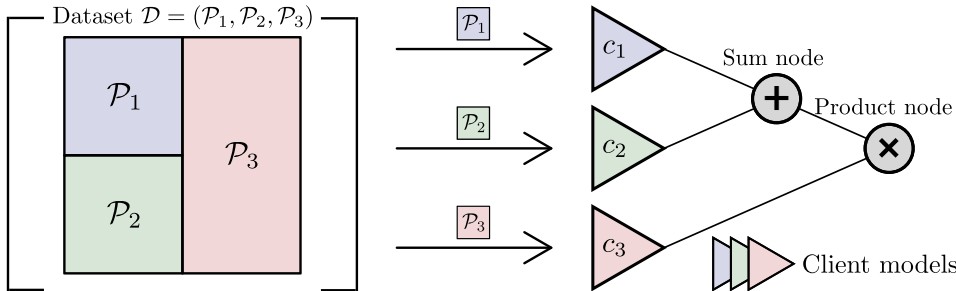

Figure 1: **Scaling PCs via Federated Circuits.** We scale PCs by splitting a dataset $\mathcal{D}$ into a set of $n$ partitions $\{\mathcal{P}_i\}_{i=1}^n$ s.t. $\mathcal{D} = \bigcup_{i=1}^n \mathcal{P}_i$. Each partition is assigned to a client (i.e., machine) $c_j$, and the resulting federated circuit (FC) is learned jointly by a set of clients. FCs represent a novel framework for federated learning (FL), capable of performing horizontal FL (samples are split across clients), vertical FL (features are split across clients), and hybrid FL (mix of horizontal and vertical).

evaluation[1] shows that FCs outperform EiNets (Peharz et al., 2020a) on large-scale density estimation tasks, demonstrating the benefits of scaling up PCs. Additionally, FCs outperform or achieve competing results on various classification tasks in all federated settings compared to state-of-the-art neural network-based and tree-based methods, demonstrating its effectiveness in FL. Overall, we make the following contributions:

**(1)** We introduce FCs, a communication-efficient and scalable FL framework unifying horizontal, vertical, and hybrid FL by mapping the semantics of PCs to FL.

**(2)** We practically instantiate FCs to FedPCs and demonstrate how the FC framework can be leveraged to scale up PCs to large real-world datasets.

**(3)** We propose a novel one-pass training scheme for FedPCs that is compatible with a broad set of learning algorithms.

**(4)** We provide extensive experiments demonstrating the effectiveness of our approach for learning large-scale PCs and performing FL. We consider multiple domains (tabular data, image data) and tasks (classification, density estimation).

We proceed as follows: After touching upon related work, we provide the probabilistic view on FL and introduce FCs. Before concluding, we present our extensive experimental evaluation of FedPCs.

## 2 PRELIMINARIES AND RELATED WORK

In the following, we briefly introduce PCs and FL and give an overview of relevant related work.

### 2.1 PROBABILISTIC CIRCUITS

PCs encode a probability distribution as a computation graph that allows for tractable inference of a wide range of queries such as conditional (partial evidence) and marginalization. Peharz et al. (2015b) define a PC over random variables $\mathbf{X}$ as a tuple $(\mathcal{G}, \phi)$ where $\mathcal{G} = (V, E)$ is a rooted, Directed Acyclic Graph (DAG) and $\phi : V \to 2^{\mathbf{X}}$ is the *scope* function assigning a subset of random variables to each node in $\mathcal{G}$. For each internal node $\mathsf{N}$ of $\mathcal{G}$ the scope is defined as the union of scopes of its children $\mathrm{ch}(\mathsf{N})$. Each leaf node $\mathsf{L}$ computes a distribution/density over its scope. All internal nodes of $\mathcal{G}$ are either a sum node $\mathsf{S}$ or a product node $\mathsf{P}$ where each sum node computes a convex combination of its children, i.e. $\mathsf{S} = \sum_{\mathsf{N} \in \mathrm{ch}(\mathsf{S})} w_{\mathsf{S},\mathsf{N}} \mathsf{N}$, and each product node computes a product of its children, i.e. $\mathsf{P} = \prod_{\mathsf{N} \in \mathrm{ch}(\mathsf{P})} \mathsf{N}$. To ensure tractability of probabilistic queries such as marginalization, a PC must be *decomposable*. Decomposability requires that for all $\mathsf{P} \in V$ it holds that $\phi(\mathsf{N}) \cap \phi(\mathsf{N}') = \emptyset$ where $\mathsf{N}, \mathsf{N}' \in \mathrm{ch}(\mathsf{P})$. To further ensure that a PC represents a valid distribution, *smoothness* must hold, i.e., for each sum $\mathsf{S} \in V$ it holds that $\phi(\mathsf{N}) = \phi(\mathsf{N}')$ where $\mathsf{N}, \mathsf{N}' \in \mathrm{ch}(\mathsf{S})$ (Peharz et al., 2015b).

---

[1] Code available at https://anonymous.4open.science/r/federated-spn-5FDC.

Decomposable and smooth PCs are often referred to as Sum-Product Networks (SPNs) (Poon & Domingos, 2011; Peharz et al., 2015a; Sánchez-Cauce et al., 2021).

Several works have tackled the goal of scaling PCs. On the architecture side, it was shown that large, random structures can be used to scale to larger problems more easily (Peharz et al., 2020b). Changes in the model layout, such as parallelizable layers and the einsum-operation (Peharz et al., 2020a) and a reduction in IO operations (Liu et al., 2024), were also shown to drastically reduce the speed of computation. Liu et al. (2022) improved the performance of PCs by latent variable distillation, where deep generative models give additional supervision during the learning process.

## 2.2 Federated Learning

In federated learning (FL), a set of data owners (or clients) aim to collaboratively learn an ML model without sharing their data. One distinguishes between horizontal, vertical, and hybrid FL depending on how data is partitioned. In horizontal FL, a dataset $\mathbf{D} \in \mathbb{R}^{n \times d}$ is partitioned s.t. each client holds the same $d$ features but different, non-overlapping sets of samples. In vertical FL, $\mathbf{D}$ is partitioned s.t. each client holds the same $n$ samples but different, non-overlapping subsets of the $d$ features. Hybrid FL describes a combination of horizontal and vertical FL where clients can hold both different (but possibly overlapping) sets of samples and features (Wen et al., 2023; Li et al., 2023a).

For all three FL settings, specifically tailored methods have been proposed to enable collaborative learning of models. The most common scheme in horizontal FL is to average the models of all clients regularly during training (McMahan et al., 2016; Karimireddy et al., 2020a;b; Sahu et al., 2018). However, model averaging requires each client to share the same model structure. In vertical FL, clients hold different feature sets; thus, there is no guarantee that the model structure can be shared among clients. In these cases, tree-based and neural models are the predominant choice and are typically learned by sharing data statistics or feature representations among clients (Kourtellis et al., 2016; Cheng et al., 2021; Vepakomma et al., 2018; Ceballos et al., 2020; Chen et al., 2020; Liu et al., 2019). Similar to tree-based vertical FL, tree-based hybrid FL approaches share data statistics (such as histograms) or model properties (such as split rules) among clients (Li et al., 2023b; 2024). However, tree-based approaches often require complex training procedures.

## 3 Federated Circuits

This work aims to scale up PCs by splitting data and the model across multiple machines, thus harnessing the availability of compute clusters to train PCs in a federated fashion. In the following, we present an elegant and effective way to achieve that using our novel federated learning framework called federated circuits (FCs). FCs unify horizontal, vertical, and hybrid FL by hierarchically learning mixtures (horizontal part) and fusing marginals (vertical part).

### 3.1 Problem Statement & Modeling Assumptions

Given a dataset $\mathbf{D}$ and a set of clients $\mathcal{C}$ where each $c \in \mathcal{C}$ holds a partition $\mathbf{D}_c$ of $\mathbf{D}$; we aim to learn the joint distribution $p(\mathbf{X})$ over random variables $\mathbf{X}$ (i.e., the features of $\mathbf{D}$). The partitioning of $\mathbf{D}$ is not further specified. Hence, each client might only hold a subset of random variables $\mathbf{X}_c \subseteq \mathbf{X}$ with support $\mathcal{X}_c$. This can be interpreted as each $c \in \mathcal{C}$ holding a dataset $\mathbf{D}_c \sim p_c$ where $p_c$ is a joint distribution over $\mathbf{X}_c$ which is related to $p(\mathbf{X})$.

We introduce two critical modeling assumptions relevant for learning a joint distribution $p(\mathbf{X})$ from a dataset $\mathbf{D}$ partitioned across a set of machines.

**Assumption 1** (Mixture Marginals). *There exists a joint distribution $p$ such that the relation $\int_{\mathbf{X} \setminus \mathbf{X}_S} p(x) = \sum_{l \in L} q(L = l) \cdot p_S(x | L = l)$ holds. Here, $\mathbf{X}_S \subseteq \mathbf{X}$ is a subset of the union of client random variables $\mathbf{X} = \cup_{c \in \mathcal{C}} \mathbf{X}_c$ with support $\mathcal{X} = \times_{c \in \mathcal{C}} \mathcal{X}_c$, each $p_S$ is defined over $\mathcal{X}_S \subseteq \mathcal{X}$ and $q$ is a prior over a latent $L$.*

To illustrate, consider a subset of variables $\mathbf{X}_S \subseteq \mathbf{X}$ shared among all clients and its complement $\mathbf{X}_{S^-} = \mathbf{X} \setminus \mathbf{X}_S$. Assumption 1 ensures that the marginal $\int_{\mathbf{X}_{S^-}} p(\mathbf{X})$ is representable as a mixture of all client distributions $p_c(\mathbf{X}_S)$ over $\mathbf{X}_S$. If Assumption 1 would not hold, the information stored on the clients' data partitions would not be sufficient to learn $p(\mathbf{X})$.

A key assumption in FL is that data cannot be exchanged among clients. However, dependencies among variables residing on different clients might still exist. To enable learning these "hidden" dependencies while keeping data private, we make the following assumption:

**Assumption 2** (Cluster Independence). *Given disjoint sets of random variables $\mathbf{X}_1, \cdots, \mathbf{X}_n$ and a joint distribution $p(\mathbf{X}_1, \cdots, \mathbf{X}_n)$, assume that a latent $L$ can be introduced s.t. the joint can be represented as $p(\mathbf{X}_1, \cdots, \mathbf{X}_n) = \sum_l p_\theta(L = l) \prod_{i=1}^n p(\mathbf{X}_i | L = l)$ where $p_\theta$ is a prior distribution over the latent $L$.*

Note that independence is only assumed within clusters in the data. Thus, the latent variable (which can be thought of as "cluster selectors") allows capturing dependencies among variables residing on different clients. Distributions of the form in Assumption 2 are strictly more expressive distribution than the product distribution and thus allow for more complex modeling:

**Fact 1.** *A joint distribution $p$ over disjoint sets of random variables $\mathbf{X}_1, \cdots, \mathbf{X}_n$ of the form $p(\mathbf{X}_1, \cdots, \mathbf{X}_n) = \sum_l p_\theta(L = l) \prod_{i=1}^n p(\mathbf{X}_i | L = l)$ is strictly more expressive than a distribution of the form $p(\mathbf{X}_1, \cdots, \mathbf{X}_n) = \prod_{i=1}^n p(\mathbf{X}_i)$. We provide proof in the App. B.*

## 3.2 BRIDGING PROBABILISTIC CIRCUITS AND FEDERATED LEARNING

We now illustrate an inherent connection between PC semantics and FL. This will allow us to train PCs on data partitioned over a set of clients and thus greatly increase the scaling potential of PCs.

**Sum Nodes and Horizontal FL.** In horizontal FL, each client is assumed to hold the same set of features, i.e., $\mathbf{X}_c = \mathbf{X}_{c'}$ for all $c, c' \in \mathcal{C}$. However, each client holds a different subset of the data. Prominent horizontal FL methods solve this task by aggregating the *model parameters* of locally learned models regularly. However, the horizontal FL setting also precisely corresponds to the interpretation of sum nodes in PCs: A sum node splits a dataset into multiple disjoint clusters. The distribution over the entire data is then represented as a mixture of the distributions learned from the disjoint clusters. Thus, instead of aggregating model parameters, we aggregate the *distributions* learned by each client on its data partition.

**Definition 1** (Horizontal FL). *Assume a set of samples $\mathbf{D}_c \sim p_c$ on each client $c \in \mathcal{C}$, a joint distribution $p$ adhering to Assumption 1 and that $\mathbf{X}_c = \mathbf{X}_{c'}$ for all $c, c' \in \mathcal{C}$ s.t. $c \neq c'$. We define horizontal FL as fitting a mixture distribution $\hat{p} = \sum_{c \in \mathcal{C}} q(c) \cdot \hat{p}_c$ such that $d(\hat{p}, p)$ and $d(p_c, \hat{p}_c)$ are minimal for all $c \in \mathcal{C}$ where $d$ is a distance metric and $\hat{p}_c$ local distribution estimates.*

This view on horizontal FL has an appealing positive side effect: Aggregating model parameters can lead to divergence during training if the client's data distributions significantly differ. We circumvent the burden of aggregating model parameters by forming a mixture of local models that can be learned independently. Thus, we do not require further assumptions on the client's distributions. Also, since clients can train models independently, the communication cost of the training is minimized.

**Product Nodes & Vertical FL.** In vertical FL, each client is assumed to hold a disjoint set of features, i.e., $\mathbf{X}_c \cap \mathbf{X}_{c'} = \emptyset$ for all $c, c' \in \mathcal{C}$. In contrast to horizontal FL, all clients hold different features belonging to the same sample instances. As in horizontal FL, there is a semantic connection between vertical FL and PCs. Product nodes in PCs compute a product distribution defined on a disjoint set of random variables. Thus, a product node separates the data along the feature dimension, corresponding to the vertical FL setting. However, a product node assumes the random variables of the child distributions to be independent of each other. Obviously, this is an unrealistic assumption for vertical FL, where features held by different clients might be statistically dependent. To capture such dependencies, Assumption 2 can be exploited, and a mixture over multiple product distributions can be formed. We will discuss this in detail in Sec. 3.3.

**Definition 2** (Vertical FL). *Assume a set of samples $\mathbf{D}_c \sim p_c$ on each data owner $c \in \mathcal{C}$, the existence of a joint distribution $p$ adhering to Assumptions 1 and 2 and that $\mathbf{X}_c \cap \mathbf{X}_{c'} = \emptyset$ holds for all $c, c' \in \mathcal{C}$ s.t. $c \neq c'$. We define vertical FL as estimating a joint distribution $\hat{p}$ s.t. $d(p, \hat{p})$ is minimal and $\int_{\mathbf{X} \setminus \mathbf{X}_c} \hat{p}(x) = \hat{p}_c(x)$ for all $x \in \mathcal{X}$ where $d$ is a distance metric and $\hat{p}_c$ are estimates of client distributions.*

**PCs & Hybrid FL.** Given Defs. 1 and 2, hybrid FL is a combination of both. In terms of PC semantics, this amounts to building a hierarchy of fusing marginals and learning mixtures. Provided with these probabilistic semantics, we can now formally bridge PCs and FL. In the following, we

distinguish between clients $\mathcal{C}$ and servers $\mathcal{S}$ and define the set of machines participating in training as $\mathcal{N} = \mathcal{C} \cup \mathcal{S}$. Bringing everything together and abstracting from the probabilistic interpretation, we define **federated circuits** (FCs) as follows.

**Definition 3** (Federated Circuits). *A **federated circuit** (FC) is a tuple $(\mathcal{G}, \psi_{\mathcal{G}}, \omega)$ where $\mathcal{G} = (V, E)$ is a rooted, Directed Acyclic Graph (DAG), $\psi_{\mathcal{G}} : V \to \mathcal{N}$ assigns each $\mathsf{N} \in V$ to a compute node $n \in \mathcal{N}$ based on the structure of $\mathcal{G}$ and $\omega : V \to O$ assigns an operation $o \in O$ to each node $\mathsf{N} \in V$ where $o : dom(\mathrm{ch}(\mathsf{N})) \to dom(\mathsf{N})$ computes the value of $\mathsf{N}$ given the values of the children of $\mathsf{N}$.*

FCs extend the definition of PCs in the sense that FCs represent a computational graph $\mathcal{G} = (V, E)$ distributed over multiple machines where arbitrary operations can be performed in each node $\mathsf{N} \in V$. Depending on the parameterization of leaves and nodes $\mathsf{N}$, FCs are not restricted to the probabilistic interpretation presented above. For example, parameterizing leaves by decision trees and introducing a node $\mathsf{N}$ that performs averaging yields a bagging model.

### 3.3 Federated Probabilistic Circuits

Let us now dive deeper into the probabilistic interpretation of FCs. To that end, we present a concrete instantiation of FCs leveraging Probabilistic Circuits (PCs) as leaf models, resulting in federated PCs (FedPCs). Following the probabilistic interpretation from Sec. 3.2, we align the PC structure with the communication network structure to form a federated PC.

**Definition 4** (Federated PC). *A Federated PC (FedPC) is a FC where each leaf node $\mathsf{C}$ is a density estimator and each node $\mathsf{N}$ s.t. $\mathrm{ch}(\mathsf{N}) \neq \emptyset$ is either a sum node ($\mathsf{S}$) or a product node ($\mathsf{P}$).*

Note that only the client nodes $\mathsf{C}$ hold a dataset and we only demand the clients to be parameterized by a density estimator. In order for FedPCs to be computationally efficient, these density estimators should be tractable. In the following, we parameterize the leaf nodes $\mathsf{C}$ as PCs.

The operation assignment $\omega$ is omitted in FedPCs as the operations performed by each node are implicitly defined (sum or product). The assignment function $\phi$ transforms the PC's computation graph into a distributed computation graph aligned to the communication network. This establishes a direct correspondence between PC semantics (computation graph) and the communication network structure in FedPCs. Inference is performed as usual in PCs by propagating likelihood values from the leaf nodes to the root node. The only difference is that the result of a node $\mathsf{N}$ has to be sent to its parent(s) $\mathbf{pa}(\mathsf{N})$ over the communication network if $\psi(\mathsf{N}) \neq \psi(\mathsf{N}')$ holds for $\mathsf{N}' \in \mathbf{pa}(\mathsf{N})$.

Training FedPCs requires adapting the regular training procedure for PCs. This is mainly because not all clients can access the same samples if data is partitioned horizontally or hybrid. Since a forward pass through a PC requires the same sample to be available on each leaf, prominent learning algorithms such as Expectation Maximization (EM) are not directly applicable in horizontal and hybrid FL settings. In the following, we propose a *one-pass* training procedure of FedPCs that does not require a full forward or backward pass over the model.

**One-Pass Training.**

Our one-pass learning algorithm learns the structure and parameters of FedPCs so that local models can be trained independently (Algo. 1, Fig. 2). Before training, all clients $c \in \mathcal{C}$ share their set of uniquely identifiable features/random variables $\mathbf{X}_c$ with a server, resulting in the feature set indicator matrix $\mathbf{M}^{|\mathcal{C}| \times |\mathbf{X}|}$ **(Lines 1-2)**. Feature identifiers can be names of features such as "account balance" and have to correspond to the same random variable on all clients (thus uniquely identifiable). Then, the server divides the joint feature space $\mathbf{X}$ into disjoint subspaces by considering all unique columns ($\mathbf{u}$) in $\mathbf{M}$. Non-unique columns indicate sets of features with cardinality $> 1$ held by multiple clients and, thus, can be modeled as a mixture in the FedPC. Hence, the subspaces $\{\mathbf{S}^{(1)}, \dots, \mathbf{S}^{(l)}\}$ represent sets of features shared by a set of clients $\{O_{\mathbf{S}^{(1)}}, \dots, O_{\mathbf{S}^{(l)}}\}$ such that the number of subspaces $l$ is minimized **(Lines 3-7)**. For example, in Fig. 1, the features of partitions 1 and 2 define one subspace as the largest subspace covering all clients holding these features (2 clients).

Afterward, the FedPC structure is constructed (bottom part of Fig. 2): First, we build a mixture (sum node) for each subspace $\mathbf{S}^{(j)}$ where $|O_{\mathbf{S}^{(j)}}| > 1$, i.e., more than one client holds $\mathbf{S}^{(j)}$ **(Lines 9-12)**. This enables each client to learn a PC over $\mathbf{S}^{(j)}$ independently. After that, $|O_{\mathbf{S}^{(j)}}| = 1$ holds for all remaining $\mathbf{S}^{(j)}$. Also, the scope of the sums nodes introduced in the FedPC share no features with any

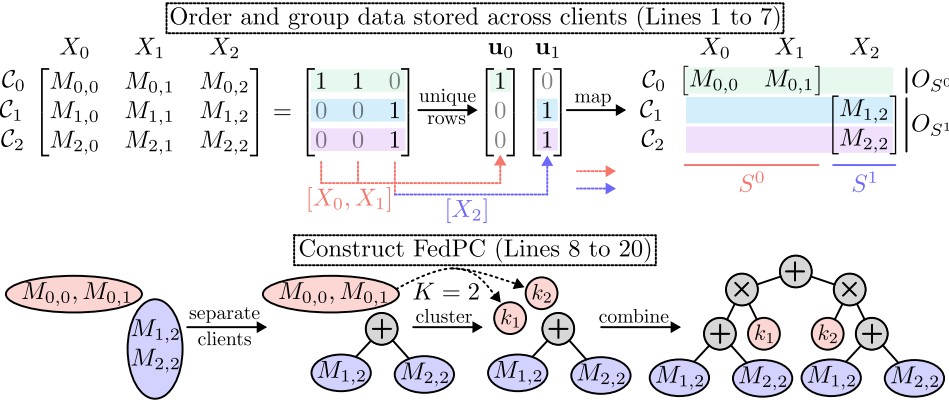

Figure 2: **One-Pass Training Visualized.** (Top) First, the matrix $\mathbf{M}$ is initialized, representing which features lie on which clusters. The unique descriptor vector $\mathbf{u}$ groups clients with the same feature subset. This forms a mapping indicating which features are available on each client. (Bottom) This mapping is utilized by first combining features that lie on different clients with sum nodes. Other features will be clustered into $K$ clusters (here $K = 2$). The final FedPC is constructed by creating product nodes containing all the sum nodes from the previous steps and at least one of the $K$ clusters. Lastly, the root node (sum node) is inserted.

of the remaining $\mathbf{S}^{(j)}$ since the server divided the feature space into disjoint subspaces. Therefore, we can use Prop. 1 and introduce $P$ product nodes to construct the remaining part of the FedPC.

To this end, we divide the data of all subspaces $\mathbf{S}^{(j)}$ where $|O_{\mathbf{S}^{(j)}}| = 1$ holds into $K$ clusters (**Line 14**). Each client learns a dedicated PC for each cluster. To ensure that the FedPC spans the entire feature space of the clients, the children of product nodes are set as follows: Each sum node introduced in the FedPC becomes a child of each product node. Additionally, for each $\mathbf{S}^{(j)}$ where $|O_{\mathbf{S}^{(j)}}| = 1$ holds, we randomly select a PC learned over one of the $K$ clusters s.t. the scope of each product node spans $\mathbf{S}$, and each PC representing a cluster is the child of at least one product node. Then, we build a mixture over all product nodes using a sum node (**Lines 15-20**). Once the FedPC is constructed, all client-sided PCs are learned. Since clients learn their PCs independently, each client can use an arbitrary learning algorithm (even different ones). As a last step, the network-sided parameters, i.e., the weights of network-sided sum nodes, of the FedPC are inferred (**Line 21-22**). For each sum node $\mathsf{S}$, the weight $\mathbf{w}_{\mathsf{S}}^{(i)}$ associated with the $i$-th child (i.e., distribution) of $\mathsf{S}$ is set to $\frac{\rho(\mathsf{N}_i)}{\sum_i \rho(\mathsf{N}_i)}$. Here, $\rho(\mathsf{N}_i) = \sum_{\mathsf{C} \in \text{ch}(\mathsf{N}_i)} |\mathbf{D}_{\mathsf{C}}|$ where $\mathbf{D}_{\mathsf{C}}$ is the dataset used to train the leaf $\mathsf{C}$. Hence, the network-sided weights can be inferred without any forward or backward pass. Note that this approach reduces horizontal FL to learning a mixture of the client's data distributions and vertical FL to learning a mixture over $P$ product nodes.

---

**Algorithm 1:** One-Pass Training

**Data:** Clients $\mathcal{C}$, features $\mathbf{X}$, cluster size $K$, FedPC fedPC

**Result:** Trained fedPC

1   $\mathbf{M} = \mathbf{0}^{|\mathcal{C}| \times |\mathbf{X}|}$;

2   $\mathbf{M}_{i,j} = 1$ if $X^{(j)}$ on client $i$;

3   map $= []$;

4   **for** $j, \mathbf{u}$ *in enum(unique_cols(* $\mathbf{M}$ *))* **do**

5     $\mathbf{S}^{(j)} = \{i : i \in \{1, \ldots, |\mathbf{X}| \wedge \text{all}(\mathbf{u} == \mathbf{M}_{:,i})\}\}$;

6     $O_{\mathbf{S}^{(j)}} = \text{argwhere}(\mathbf{u} == 1)$;

7     map.append$(\mathbf{S}^{(j)}, O_{\mathbf{S}^{(j)}})$;

8   sums $= []$;

9   **for** $\mathbf{S}^{(j)}, O_{\mathbf{S}^{(j)}}$ *in map* **do**

10    **if** $|O_{\mathbf{S}^{(j)}}| > 1$ **then**

11      s = fedPC.add_sum$(\mathbf{S}^{(j)}, O_{\mathbf{S}^{(j)}})$;

12      sums.add(s)

13    **else**

14      client_clusters = cluster_local_data$(O_{\mathbf{S}^{(j)}}, K)$;

15   products = fedPC.add_products$(P)$;

16   **for** *prod in products* **do**

17    prod.children.add(sums);

18    **for** *client, clusters in client_clusters* **do**

19      prod.children.add_rand_subset(clusters);

20   fedPC.add_mixture_over_products(products);

21   fedPC.train_clients();

22   fedPC.infer_weights();

23   **return** fedPC

---

| | Log-Likelihood | | | | Relative Runtime | | | |
|---|---|---|---|---|---|---|---|---|
| | cent | horizontal | vertical | hybrid | cent | horizontal | vertical | hybrid |
| MNIST | $3352_{\pm 3.5}$ | $3350_{\pm 3.2}$ | $3351_{\pm 3.8}$ | $3349_{\pm 3.7}$ | 1.0 | $\mathbf{0.07}_{\pm \mathbf{0.01}}$ | $0.13_{\pm 0.01}$ | $0.13_{\pm 0.02}$ |
| Income | $-11.5_{\pm 0.1}$ | $-11.4_{\pm 3.5}$ | $-11.9_{\pm 3.3}$ | $-12.0_{\pm 1.5}$ | 1.0 | $\mathbf{0.17}_{\pm \mathbf{0.02}}$ | $0.236_{\pm 0.01}$ | $0.21_{\pm 0.02}$ |
| Cancer | $-38.9_{\pm 0.3}$ | $-38.5_{\pm 1.1}$ | $-38.6_{\pm 0.5}$ | $-38.7_{\pm 1.5}$ | 1.0 | $\mathbf{0.21}_{\pm \mathbf{0.07}}$ | $0.35_{\pm 0.05}$ | $0.35_{\pm 0.1}$ |
| Credit | $-12.8_{\pm 1.0}$ | $-13.1_{\pm 0.5}$ | $-12.5_{\pm 2.3}$ | $-12.5_{\pm 1.3}$ | 1.0 | $0.42_{\pm 0.05}$ | $\mathbf{0.31}_{\pm \mathbf{0.09}}$ | $0.40_{\pm 0.13}$ |

Table 1: **FedPCs speed up training while retaining model performance.** We trained PCs in a centralized setting (cent.) and in all FL settings (using FedPCs) on different datasets and the same structure learning algorithm. We find that FedPCs tremendously speed up training (reported as relative runtime w.r.t. centralized training where relative centralized runtime is 1.0 while there is no reduction in log-likelihood. This demonstrates that PCs can be learned in federated settings (positive log-likelihoods due to Gaussian leaves).

Next, we analyze the communication efficiency of our proposed learning algorithm.

### 3.4 ANALYSIS OF COMMUNICATION EFFICIENCY

Communication efficiency is a key requirement for efficient training when learning models on scale on partitioned data, such as in FL. We now analyze the communication efficiency of FedPCs.

**Horizontal FL.** Assume a client set $\mathcal{C}$ where each client holds a model with $M$ parameters. Further, assume models are aggregated $K$ times during training ($K$ communication rounds). Then, model aggregation-based algorithms like FedAvg commonly used in horizontal FL send $\mathcal{O}(M \cdot |\mathcal{C}| \cdot K)$ messages over the network as each client sends $M$ model parameters to a server in each communication round. Training FedPCs with one-pass training, in contrast, only requires $\mathcal{O}(|\mathcal{C}| \cdot (M + 1))$ messages over the network as models are learned locally and independently of each other, followed by setting the parameters ($\mathcal{O}(|\mathcal{C}|)$ messages) of the sum nodes and aggregating the model on the server ($\mathcal{O}(M|\mathcal{C}|)$ messages).

**Vertical FL.** In vertical settings, SplitNN-like architectures are commonly used. Assume training a SplitNN architecture for $E$ epochs that output a feature vector of size $F$ for each sample of a dataset with $S$ samples, vertically distributed over clients $\mathcal{C}$. The training requires sending $\mathcal{O}(E \cdot |\mathcal{C}| \cdot F \cdot S)$ messages over the network. In contrast, with one-pass training of FedPCs, each client learns a dedicated PC with $M$ parameters for each of the $K$ clusters that are learned. The last layer of the FedPC is a mixture of $P$ products of clusters. The mixture parameters are set after training each client's model. Aggregating the learned models and setting the network-sided mixture parameters requires $\mathcal{O}(K \cdot M \cdot |\mathcal{C}| + P)$ messages to be sent. If $(K \cdot M + \frac{P}{|\mathcal{C}|}) < (E \cdot F \cdot S)$ holds, training FedPCs is more communication efficient than training SplitNN-like architectures. In practice, this is likely to hold: The number of clusters is usually smaller than 100 while feature vectors can have hundreds of dimensions (i.e., $F > 100$). Further, models should have fewer parameters than samples in the dataset to ensure generalization (i.e., $M < S$). $P$ can be set to an arbitrary value, depending on $|\mathcal{C}|$ and the data. App. E provides more details and an intuition on communication costs.

**Hybrid FL.** In hybrid FL, FedPCs are trained on several subspaces: There are subspaces present on all or a subset of clients (denoted as $R_s$) and there are subspaces only available on one client (denoted as $R_d$). Further denote communication costs of FedPCs in horizontal FL and vertical FL as $C_h$ and $C_v$, respectively. Since the training procedure in hybrid cases essentially performs horizontal FL on shared feature spaces and vertical FL on disjoint feature spaces, $\mathcal{O}(|R_s| \cdot C_h + |R_v| \cdot C_v)$ messages are sent over the network during training.

**Remark 1.** *When scaling PCs using FedPCs, we do not aggregate the models after training. This distributes computation load across multiple machines also during inference and further decreases communication costs during training.*

## 4 EXPERIMENTS

In our empirical evaluation, we corroborate that FedPCs can be leveraged to effectively scale up PCs via data and model

partitioning. By performing horizontal, vertical and hybrid FL in one unified framework, we obtain high-performing models with the same or improved performance compared to prominent FL baselines. We aim to answer the following questions: **(Q1)** Can FedPCs decrease the required training time and successfully learn a joint distribution over distributed data? **(Q2)** Do FedPCs effectively scale up PCs, thus yielding more expressive models? **(Q3)** How do FCs with different parameterizations perform on classification tasks compared to existing FL methods? **(Q4)** How does our one-pass learning algorithm compare to training with the EM algorithm?

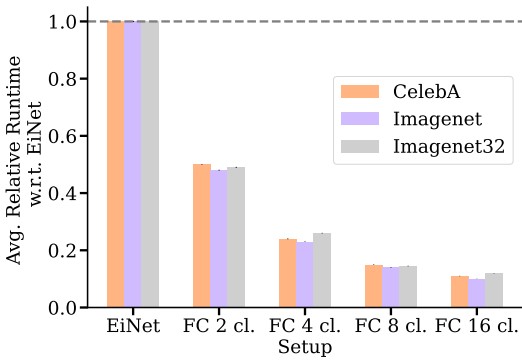

Figure 3: **FedPCs speed up training.** Due to parallel training on multiple, separate data partitions, FedPCs tremendously speed up training compared to EiNet (shown in relative speed-up).

**Experimental Setup.** To see if FedPCs, an instantiation of FCs, successfully scale up PCs, we follow Liu et al. (2024) and perform density estimation on three large-scale, high-resolution image datasets: Imagenet, Imagenet32 (both 1.2M samples), and CelebA (200K samples). The datasets were partitioned over 2-16 clients horizontally. We compare FedPCs to EiNets and Pyjuice.

To evaluate FCs in FL scenarios, we selected three tabular datasets that cover various application domains and data regimes present in the real world: one credit fraud dataset ($\sim$ 300K samples), a medical dataset (breast cancer detection; $<$ 1000 samples), and the popular Income dataset ($>$ 1M samples). The selected datasets for FL cover low-data, medium-data, and large-data regimes[2]. Both balanced (breast cancer) and imbalanced (income, credit) datasets are included in our evaluation. We selected tabular datasets as they are well suited to investigate FCs in horizontal, vertical, and hybrid settings and represent various real-world applications. We compare FCs to FedAvg (horizontal) and SplitNN (vertical), both using TabNet (Arik & Pfister, 2020) as neural network architecture parameterization. Additionally, we compare FCs to FedTree (Li et al., 2023b). For more details on the experimental protocol, see App. F.

**(Q1) FedPCs learn joint distributions over partitioned data in less time.** First, we validate that FedPCs correctly and efficiently perform density estimation on partitioned datasets distributed over multiple clients. To this end, multiple tabular datasets were distributed over a set of clients corresponding to horizontal (5 clients), vertical (2 clients), and hybrid FL (2 clients). To demonstrate that FedPCs are also robust against label shifts, a common regime in FL, each client received data from only a subset of classes in the horizontal case, and local PCs were learned over the client samples. In the vertical case, we split data s.t. feature spaces of clients are disjoint, but each client holds the same samples. In hybrid settings, data was distributed s.t. both feature- and sample-spaces among clients have overlaps (but no full overlap). For all tabular datasets, the leaves of the FedPC were parameterized with MSPNs (Molina et al., 2018), a member of the PC model family that is capable of performing density estimation on mixed data domains (i.e., continuous as well as discrete random variables). We chose MSPNs as the centralized models, which were learned using LEARNSPN, a recursive greedy structure learning algorithm for SPNs Gens & Domingos (2013). For MNIST, EiNets with Gaussian densities were used as PC instantiations in all settings.

Tab. 1 compares log-likelihood scores and relative runtime of centralized training of a PC on the full datasets with log-likelihood scores and relative runtimes achieved by FedPC in different FL settings. FedPCs successfully reproduce the results of centralized PCs on tabular datasets while being tremendously faster in training. This validates our approach and we answer **(Q1)** affirmatively.

**(Q2) FedPCs effectively scale up PCs.** To examine whether FedPCs can be leveraged to scale up PCs effectively, we trained an EiNet, PyJuice, and FedPC on CelebA, Imagenet32, and Imagenet. All models used the Poon-Domingos (PD) architecture. FedPCs were parameterized with EiNets, and data was distributed among 2, 4, 8, and 16 clients. The FedPC model and baseline models (EiNets and PyJuice) were selected to ensure that each fits within a single GPU (see App. F for system details). All models were parameterized with Gaussian leaves. Before training, data was clustered

---

[2]see App. F for more details

|  | CelebA | Imagenet32 | Imagenet |
|---|---|---|---|
| EiNet (Peharz et al., 2020a) | $5842.62 \pm 94.9$ | $682.82 \pm 3.50$ | $-5893.59 \pm 84.79$ |
| PyJuice (Liu et al., 2024) | $4228.14 \pm 25.5$ | $664.54 \pm 6.41$ | $-5732.21 \pm 71.25$ |
| FedPC (2 clients) | $\mathbf{6337.50 \pm 98.3}$ | $1044.38 \pm 8.02$ | $-4971.36 \pm 120.83$ |
| FedPC (4 clients) | $\underline{6279.98 \pm 86.9}$ | $1196.39 \pm 1.50$ | $-2330.87 \pm 162.17$ |
| FedPC (8 clients) | $6019.53 \pm 96.3$ | $\mathbf{1205.05 \pm 2.72}$ | $\underline{-1818.17 \pm 81.12}$ |
| FedPC (16 clients) | $5387.62 \pm 82.9$ | $\underline{1197.69 \pm 10.79}$ | $\mathbf{-1157.23 \pm 74.29}$ |

Table 3: **FedPCs outperform EiNets and PyJuice on density estimation tasks.** FedPCs achieve better results on density estimation tasks on three challenging image datasets (CelebA, Imagenet32 and Imagenet). This is because FedPCs can learn far larger models distributed across multiple machines. Results are reported as log-likelihood values (higher is better). Note that we used Gaussian *densities* as PC leaves; thus, log-likelihood can get positive. Best value in **bold**, 2nd best underlined.

on encodings of a pre-trained Vision Transformer (Dosovitskiy et al., 2021), and the images were distributed horizontally, s.t. each client holds approximately equally large clusters. To ensure a fair comparison, EiNets and PyJuice were trained using the same clusters. The leafs and all baselines were trained with EM. In Tab. 3, we show the log-likelihood values achieved by EiNets, PyJuice, and FedPC computed over the same test set. For Imagenet and Imagenet32, log-likelihood improves with an increasing number of participating clients.

|  | EM | one-pass |
|---|---|---|
| Synth. Data | $-53.6 \pm 1.3$ | $-53.2 \pm 1.2$ |
| Income | $-18.5 \pm 0.1$ | $-18.0 \pm 0.5$ |
| Breast-Cancer | $-52.3 \pm 0.2$ | $-55.7 \pm 0.2$ |
| Credit | $-26.7 \pm 1.2$ | $-28.3 \pm 0.4$ |

Table 2: **One-pass training retains performance.** We trained the same FedPC architecture on various datasets using EM and one-pass training in a vertical setting. The average log-likelihood value of the hold-out test set across 10 runs is reported.

On CelebA, log-likelihood increases when we scale up to two participating clients. For 8 and 16 clients, the log-likelihood decreases again. We posit that this is because CelebA consists of a low number of relatively homogeneous clusters. Thus, increasing the cluster and model size to 8/16 could lead to overfitting and thus decreasing log-likelihoods. Since Imagenet consists of much more heterogeneous images, larger models and a larger number of clusters are beneficial for learning (see App. D for more details). Additionally using a larger number of clients reduces training time significantly (see Fig. 3). FedPCs thus efficiently scale tractable probabilistic models to large datasets.

**(Q3) FCs achieve state of the art classification results in FL.** FCs can be parameterized with different models in the leaves. We examine two parameterizations to solve a federated classification task on three tabular datasets. First, we use the FedPC (FC [PC]) from **(Q1)**, which can be used to solve discriminative tasks leveraging tractable computation of conditionals in PCs. The second FC parameterization we examine is decision trees (FC [DT]), representing an instantiation of a bagging model. To see how FCs perform in federated classification tasks, we compare FCs to well-known methods for horizontal FL and vertical FL. The experiments were conducted on tabular datasets covering various real-world application domains and distribution properties. We employ TabNet and FedTree as strong baselines. In the horizontal FL setting, TabNet was trained using FedAvg; in the vertical FL setting, it was trained in a SplitNN fashion (Ceballos et al., 2020). The results were compared against our one-pass training. FCs yield comparable or even better results than the selected baselines on all datasets (see Fig 4; App. D) while being significantly more flexible since FCs can be trained with the same unified procedure in all FL settings. In contrast, training neural networks requires substantial changes to the training procedure once the FL setting switches. Hence, FCs are more flexible while still competitive or better than prominent FL baselines.

**(Q4) One-pass training retains performance.** To see how the proposed one-pass training compares to training PCs with standard optimization algorithms such as EM, we define an FL setup where data exchange is allowed. This is necessary as we have to train the PC and FedPC architecture with EM to compare to our one-pass procedure. We used RAT-SPNs (Peharz et al., 2020b) as leaf parameterizations of the FedPC. Then, we trained a FedPC using standard EM (i.e., data exchange was allowed) and another FedPC with the same FedPC architecture on a vertically split dataset using our one-pass procedure. We report the final average log-likelihood of the test dataset, both for EM training and one-pass training (see Tab. 2). It can be seen that there is no significant decrease in

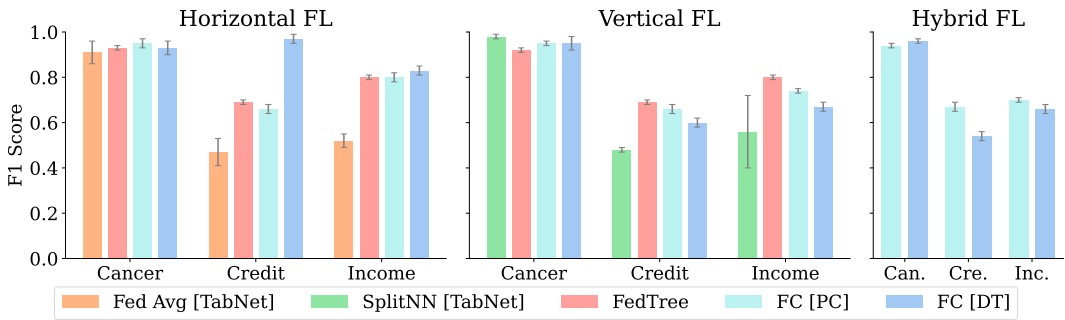

Figure 4: **FCs are competitive to prominent FL methods in all settings.** FCs achieve competitive performance on various classification tasks compared to prominent horizontal/vertical FL baselines. FCs also handle the more challenging setting of hybrid FL without performance drops. We reported the F1 score as we consider binary classification tasks with imbalanced datasets.

log-likelihood in any case. Hence, our results indicate that one-pass training is preferable since it is communication efficient.

## 5 CONCLUSION

In this work, we introduced federated circuits that hinge on an inherent connection between PCs and FL. We demonstrated that both the training speed and expressivity of PCs can be increased by learning PCs on scale across partitioned data. Since our framework allows for the integration of various types of density estimators, other models and advances of PCs and other fields can be integrated seamlessly, maintaining the relevance of the federated approach for scaling.

**Limitations and Future Work.** While our experiments showed that scaling PCs can considerably improve training speed and performance, scaling to such large-scale models requires sufficient computational resources. For future work, investigating other parametrizations for FCs beyond PCs is promising. Additionally, it is interesting how the probabilistic framework for hybrid FL could also benefit more traditional FL applications, apart from scaling PCs.

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

## A  NOTATION

The following table provides an overview of all symbols used throughout the paper, each with a brief description.

| Symbol | Meaning |
|---|---|
| $\mathbf{X}$ | Set of random variables |
| $\mathbf{X}_c$ | Set of random variables on client c |
| $\mathbf{D}$ | Dataset |
| $\mathbf{D}_c$ | Dataset on client c |
| $\mathcal{C}$ | set of clients |
| $p$ | joint distribution |
| $p_c$ | marginal distribution over all random variables held by client c |
| $\hat{p}$ | distribution from data |
| N | node in PC/FC |
| C | client node in FC |
| S, P | Sum/Product node in PC/FC |
| $\psi$ | scope function in PC/FC |
| $\omega$ | function assigning compute nodes to nodes of FC. Defines alignment between FC structure and communication network. |

## B  PROOFS

In this section we give full proofs for our propositions in the paper.

### B.1  FACT 1

A joint distribution $p$ over disjoint sets of random variables $\mathbf{X}_1, \cdots, \mathbf{X}_c$ of the form $p(\mathbf{X}_1, \cdots, \mathbf{X}_c) = \sum_l p_\theta(L = l) \prod_k^c p(\mathbf{X}_k | L = l)$ is strictly more expressive than a distribution of the form $p(\mathbf{X}_1, \cdots, \mathbf{X}_c) = \prod_k^c p(\mathbf{X}_k)$.

*Proof.* We have to prove two things here: (1) A mixture consisting of one component equals the product distribution for the distribution family assumed in Proposition 1 and (2) a latent variable model is strictly more expressive than the product distribution.

**(1):** For a latent $L$ with $|\text{supp}\{L\}| = 1$ (hence $p(L)$ is a point mass), $\sum_l p_\theta(L = l) \prod_{k=1}^c p(\mathbf{X}_k | L = l) = \prod_{k=1}^c p(\mathbf{X}_k)$ holds as for $p_\theta(L = l) = 1$ for the only $l \in \text{supp}\{L\}$. Also, if there is only one mixture component, conditioning on the only component has no effect, i.e. $p(\mathbf{X}_k | L = l) = p(\mathbf{X}_k)$.

**(2):** Assume an $n$-dimensional space $\mathcal{X}_k = \mathcal{X}_{k_1} \times \cdots \times \mathcal{X}_{k_n}$ for each set of variables $\mathbf{X}_k$ and a $c \times n \times m$ tensor $\mathbb{X}$ of random variables where each $\mathbb{X}_k$ corresponds to a matrix/set of random variables $\mathbf{X}_k = (X_{11}, \ldots, X_{nm})$, i.e. there exist $m$ random variables per dimension of $\mathcal{X}_k$. Further assume a distribution $p_{\theta_{kij}}$ for each $\mathbb{X}_{kij}$ parameterized by $\theta_{kij}$ and that $\mathbb{X}_{kij} \perp\!\!\!\perp \mathbb{X}_{k'lj}$ holds for all $k \neq k'$ and $l \neq i$. Note that this does not forbid dependencies among variables within each matrix $\mathbb{X}_k$. Due to our independence assumption we can define distributions $p_{\theta_j} = \prod_{k=1}^c p(\mathbb{X}_{k:j})$ for each $j$. Since each of these distributions is defined over $\mathcal{X}$, we can introduce a latent $L$ with support $\{1, \ldots, m\}$ and associated prior $p_\theta(L)$, yielding a mixture of $c$ components over vectorized random variables. Hence we can write $p(\mathbb{X}) = \sum_{l=1}^C p_\theta(L = l) \cdot p(\mathbb{X} | L = l)$. This can be rewritten as $p(\mathbb{X}) = \sum_{l=1}^c p_\theta(L = l) \cdot p(\mathbb{X}_l)$. As each $p(\mathbb{X}_l)$ is a product distribution over random variables corresponding to some mixture component $j$, rewriting yields $p(\mathbb{X}) = \sum_{l=1}^c p_\theta(L = l) \cdot \prod_{j=1}^c p(\mathbb{X}_{l:j})$. Using (1), setting $|\text{supp}\{L\}| = 1$ and setting the number of mixtures also to 1 yields a special case, namely the product distribution over the only defined mixture component $j$, i.e. $\prod_j p(\mathbb{X}_{l:j})$. Hence a mixture as we have defined it is strictly more expressive as a single product distribution. $\square$

### B.2 PROPOSITION 2

Assumption 2 aligns with the principle of maximum entropy: we aim to find the joint distribution with maximum entropy *within* clusters while allowing for dependencies among clients' random variables and ensuring the marginals for each client are preserved. Although multiple joint distributions can preserve the marginals, non-maximal entropy solutions introduce additional assumptions or prior knowledge, limiting flexibility. By assuming independence of all variables within a cluster, we efficiently construct the maximum entropy distribution via a mixture of product distributions. For independent variables, the product distribution maximizes entropy, as can be shown by leveraging the joint and conditional differential entropy. Given random variables $\mathbf{X} = X_1, \ldots, X_n$ and a density $p$ defined over support $\mathcal{X} = \mathcal{X}_1 \times \cdots \times \mathcal{X}_n$, the joint differential entropy is defined as:

$$h(\mathbf{X}) = \int_{\mathcal{X}} p(x_1, \ldots, x_n) \log p(x_1, \ldots, x_n) \tag{1}$$

The conditional differential entropy for two sets of random variables $\mathbf{X}$ and $\mathbf{Y}$ and a joint distribution $p(\mathbf{X}, \mathbf{Y})$ defined over support $\mathcal{X} \times \mathcal{Y}$ is defined analogously:

$$h(\mathbf{X}|\mathbf{Y}) = \int_{\mathcal{X}, \mathcal{Y}} p(\mathbf{x}, \mathbf{y}) \log p(\mathbf{x}|\mathbf{y}) \tag{2}$$

Given two sets of random variables $\mathbf{X}, \mathbf{Y}$ with densities $p(\mathbf{X})$ and $p(\mathbf{Y})$ and support $\mathcal{X}, \mathcal{Y}$ respectively, the joint $p(\mathbf{X}, \mathbf{Y}) = p(\mathbf{X}) \cdot p(\mathbf{Y})$ is the maximum entropy distribution if $\mathbf{X}$ and $\mathbf{Y}$ are mutually independent.

*Proof.* We consider the two cases that $\mathbf{X}$ and $\mathbf{Y}$ are mutually independent and that they are not mutually independent. The joint entropy can be written as $h(\mathbf{X}, \mathbf{Y}) = h(\mathbf{X}|\mathbf{Y}) + h(\mathbf{Y})$. In the case of mutual independence, this reduces to $h(\mathbf{X}, \mathbf{Y}) = h(\mathbf{X}) + h(\mathbf{Y})$. Hence it has to be shown that $h(\mathbf{X}|\mathbf{Y}) < h(\mathbf{X})$ holds if $\mathbf{X}$ and $\mathbf{Y}$ are not mutually independent:

$$h(\mathbf{X}|\mathbf{Y}) < h(\mathbf{X})$$

$$\equiv -\int_{\mathcal{X}, \mathcal{Y}} p(\mathbf{x}, \mathbf{y}) \log p(\mathbf{x}|\mathbf{y}) < -\int_{\mathcal{X}, \mathcal{Y}} p(\mathbf{x}, \mathbf{y}) \log p(\mathbf{x})$$

$$\equiv -\left( \int_{\mathcal{X}, \mathcal{Y}} p(\mathbf{x}, \mathbf{y}) \log p(\mathbf{x}|\mathbf{y}) - \int_{\mathcal{X}, \mathcal{Y}} p(\mathbf{x}, \mathbf{y}) \log p(\mathbf{x}) \right) < 0$$

$$\equiv -\left( \int_{\mathcal{X}, \mathcal{Y}} p(\mathbf{x}, \mathbf{y}) \log \frac{p(\mathbf{x}|\mathbf{y})}{p(\mathbf{x})} \right) < 0$$

Since $\mathbf{X} \perp\!\!\!\perp \mathbf{Y}$ holds where $\perp\!\!\!\perp$ means mutual independence, $\frac{p(\mathbf{x}|\mathbf{y})}{p(\mathbf{x})} \neq 1$ at least for some $\mathbf{x}, \mathbf{y}$. Since the mutual independence $I(\mathbf{X}, \mathbf{Y}) = \int_{\mathcal{X}, \mathcal{Y}} p(\mathbf{x}, \mathbf{y}) \log \frac{p(\mathbf{x}, \mathbf{y})}{p(\mathbf{x}) \cdot p(\mathbf{y})}$ can be represented as $I(\mathbf{X}, \mathbf{Y}) = h(\mathbf{X}) - h(\mathbf{X}|\mathbf{Y})$, $I(\mathbf{X}, \mathbf{Y}) \geq 0$ holds and $-\left( \int_{\mathcal{X}, \mathcal{Y}} p(\mathbf{x}, \mathbf{y}) \log \frac{p(\mathbf{x}|\mathbf{y})}{p(\mathbf{x})} \right) = h(\mathbf{X}|\mathbf{Y}) - h(\mathbf{X})$ it follows that $h(\mathbf{X}) > h(\mathbf{X}|\mathbf{Y})$.

$\square$

## C ALGORITHMS

In this section we provide pseudo-code for the end-to-end training algorithm, the two-step training algorithm and the FedSPN structure construction in hybrid FL scenarios.

### C.1 EM TRAINING

In vertical FL settings, a full forward and backward pass can be computed in FedPCs. Thus, we provide a distributed EM training algorithm here.

---

**Algorithm 2:** EM Training

---

**Data:** FedPC-parameter tuple $\langle s, p \rangle$
**Data:** Distributed Dataset $\mathbf{D}$
**Result:** Trained FedPC $s$

1   $g \leftarrow 0$
2   **for** *random batch* $\mathbf{x}$ *from* $\mathbf{D}$ **do**
3      $\ell \leftarrow \log(s(\mathbf{x}))$
4      $\nabla_p s(\mathbf{x}) \leftarrow \text{distributed\_backward}(\ell, \mathbf{x}, s, p)$
5      $\text{em\_step}(p, \nabla_p s(\mathbf{x}))$

---

---

**Algorithm 3:** Distributed Backward

---

**Data:** FedPC-parameter tuple $\langle s, p \rangle$
**Data:** Batch $\mathbf{x}$
**Data:** Log-likelihood $\ell$
**Result:** Trained FedPC $s$

1   $g \leftarrow 0$
2   $\text{gradients} \leftarrow []$
3   **for** *sum node* $\mathsf{S} \in s$ **do**
4      $g_{\text{pa}(\mathsf{S})} \leftarrow []$
5      **for** $N \in \text{pa}(\mathsf{S})$ **do**
6         **if** $N \notin \phi(\mathsf{S})$ **then**
7            obtain $\nabla_N(\mathbf{x})\ell$ from $\phi(\mathsf{S})$
8         **else**
9            compute $\nabla_N(\mathbf{x})\ell$
10         add $\nabla_N(\mathbf{x})\ell$ to $g_{\text{pa}(\mathsf{S})}$
11      compute $g_{p(\mathsf{S})} \leftarrow \sum_{g \in g_{\text{pa}(\mathsf{S})}} \nabla_{p(\mathsf{S})} \sum_{c \in \text{ch}(\mathsf{S})} p_c(\mathsf{S})c(\mathbf{x})$
12      add $\langle p(\mathsf{S}), g_{p(\mathsf{S})} \rangle$ to gradients
13   **return** gradients

---

## D   FURTHER RESULTS

Here, we provide further experimental details on FCs.

**Model Parameter Ablation.** To validate our results, we provide an additional ablation study on the effect the model size (measured in the number of parameters) has on the final model performance. To this end, we trained models of different sizes (1.2M, 34M, and 99M parameters) on CelebA. We used equally clustered data (2, 4, 8, or 16 clusters) and trained a mixture of EiNets in each run to ensure that no other effects affect the result. We find that the model parameters have a significant effect on the final model performance (reported as log-likelihood) and larger models achieve better log-likelihood values. Thus, our ablation confirms that scaling PCs is crucial to obtaining high-quality density estimates on complex data. For detailed results, see Tab. 4.

|  | 2 clusters | 4 clusters | 8 clusters | 16 clusters |
|---|---|---|---|---|
| 1.2M param. | $-3692.40 \pm 67.07$ | $-3263.54 \pm 102.60$ | $-3668.98 \pm 87.66$ | $-5145.27 \pm 64.28$ |
| 34M param. | $1659.57 \pm 65.02$ | $1154.19 \pm 55.31$ | $481.02 \pm 103.37$ | $-1104.55 \pm 109.69$ |
| 99M param. | $\mathbf{5011.55 \pm 95.57}$ | $\mathbf{4388.37 \pm 67.94}$ | $\mathbf{3727.43 \pm 71.29}$ | $\mathbf{2208.78 \pm 38.82}$ |

Table 4: **Model size significantly influences log-likelihood.** We trained mixtures of EiNets of various sizes on the same clustering of CelebA to validate our results from the main paper. The model size has a crucial influence on the final model performance and larger models achieve better log-likelihoods.

**FL Classification Results.** We compare FCs to several baselines in horizontal, vertical, and hybrid FL. In horizontal FL, we compare against FedAvg (using TabNet (Arik & Pfister, 2020)) and FedTree (Li et al., 2023b); in vertical FL, we compare against SplitNN (also using TabNet) and FedTree. In hybrid

FL, we compare different parameterizations of FCs (FedPCs and FCs parameterized with decision trees). We find that FCs are competitive or outperforming the selected baselines in all FL settings (see Tab. 5). This makes them a very flexible FL framework that still yields high-performing models.

| | | Cancer | | Credit | | Income | |
|---|---|---|---|---|---|---|---|
| | | Acc. | F1 | Acc. | F1 | Acc. | F1 |
| Horizontal FL | FedAvg [TabNet] (5 cl.) | $0.92 \pm 0.03$ | $0.92 \pm 0.03$ | $0.71 \pm 0.11$ | $0.48 \pm 0.04$ | $0.68 \pm 0.06$ | $0.51 \pm 0.03$ |
| | FedAvg [TabNet] (10 cl.) | $0.92 \pm 0.04$ | $0.91 \pm 0.05$ | $0.56 \pm 0.12$ | $0.47 \pm 0.06$ | $0.64 \pm 0.06$ | $0.52 \pm 0.03$ |
| | FedTree (5 cl.) | $0.93 \pm 0.01$ | $0.92 \pm 0.01$ | $0.91 \pm 0.01$ | $0.63 \pm 0.01$ | $0.88 \pm 0.01$ | $0.82 \pm 0.02$ |
| | FedTree (10 cl.) | $0.94 \pm 0.01$ | $0.93 \pm 0.01$ | $0.92 \pm 0.01$ | $0.69 \pm 0.01$ | $0.87 \pm 0.01$ | $0.80 \pm 0.01$ |
| | FC [PC] (5 cl.) | $0.98 \pm 0.01$ | $0.98 \pm 0.01$ | $0.93 \pm 0.02$ | $0.68 \pm 0.02$ | $0.87 \pm 0.02$ | $0.80 \pm 0.01$ |
| | FC [PC] (10 cl.) | $0.95 \pm 0.02$ | $0.95 \pm 0.02$ | $0.93 \pm 0.01$ | $0.66 \pm 0.02$ | $0.87 \pm 0.01$ | $0.80 \pm 0.02$ |
| | FC [DT] (5 cl.) | $0.95 \pm 0.03$ | $0.93 \pm 0.02$ | $0.92 \pm 0.01$ | $0.67 \pm 0.01$ | $0.89 \pm 0.01$ | $0.83 \pm 0.01$ |
| | FC [DT] (10 cl.) | $0.95 \pm 0.02$ | $0.93 \pm 0.03$ | $0.92 \pm 0.01$ | $0.97 \pm 0.02$ | $0.89 \pm 0.01$ | $0.83 \pm 0.02$ |
| | SplitNN [TabNet] | - | - | - | - | - | - |
| Vertical FL | SplitNN [TabNet] (2 cl.) | $0.98 \pm 0.01$ | $0.98 \pm 0.01$ | $0.93 \pm 0.01$ | $0.48 \pm 0.01$ | $0.56 \pm 0.25$ | $0.42 \pm 0.17$ |
| | SplitNN [TabNet] (3 cl.) | $0.98 \pm 0.01$ | $0.98 \pm 0.01$ | $0.93 \pm 0.01$ | $0.48 \pm 0.01$ | $0.62 \pm 0.20$ | $0.56 \pm 0.16$ |
| | FedTree (2 cl.) | $0.94 \pm 0.01$ | $0.93 \pm 0.01$ | $0.92 \pm 0.01$ | $0.69 \pm 0.02$ | $0.87 \pm 0.01$ | $0.80 \pm 0.01$ |
| | FedTree (3 cl.) | $0.93 \pm 0.01$ | $0.92 \pm 0.01$ | $0.92 \pm 0.01$ | $0.69 \pm 0.01$ | $0.87 \pm 0.01$ | $0.80 \pm 0.01$ |
| | FC [PC] (2 cl.) | $0.96 \pm 0.01$ | $0.96 \pm 0.01$ | $0.92 \pm 0.01$ | $0.67 \pm 0.01$ | $0.84 \pm 0.02$ | $0.74 \pm 0.01$ |
| | FC [PC] (3 cl.) | $0.95 \pm 0.01$ | $0.95 \pm 0.01$ | $0.92 \pm 0.01$ | $0.66 \pm 0.02$ | $0.84 \pm 0.01$ | $0.74 \pm 0.01$ |
| | FC [DT] (2 cl.) | $0.96 \pm 0.01$ | $0.96 \pm 0.02$ | $0.93 \pm 0.01$ | $0.60 \pm 0.02$ | $0.83 \pm 0.02$ | $0.67 \pm 0.02$ |
| | FC [DT] (3 cl.) | $0.95 \pm 0.01$ | $0.95 \pm 0.03$ | $0.93 \pm 0.01$ | $0.60 \pm 0.02$ | $0.82 \pm 0.02$ | $0.67 \pm 0.02$ |
| | FedAvg [TabNet] | - | - | - | - | - | - |
| Hybrid FL | FC [PC] (2 cl.) | $0.94 \pm 0.01$ | $0.94 \pm 0.01$ | $0.92 \pm 0.01$ | $0.67 \pm 0.01$ | $0.82 \pm 0.02$ | $0.71 \pm 0.01$ |
| | FC [PC] (3 cl.) | $0.94 \pm 0.01$ | $0.94 \pm 0.01$ | $0.92 \pm 0.01$ | $0.67 \pm 0.02$ | $0.80 \pm 0.01$ | $0.70 \pm 0.01$ |
| | FC [DT] (2 cl.) | $0.96 \pm 0.01$ | $0.96 \pm 0.02$ | $0.93 \pm 0.01$ | $0.60 \pm 0.02$ | $0.82 \pm 0.02$ | $0.66 \pm 0.02$ |
| | FC [DT] (3 cl.) | $0.96 \pm 0.01$ | $0.96 \pm 0.01$ | $0.93 \pm 0.01$ | $0.54 \pm 0.02$ | $0.82 \pm 0.02$ | $0.66 \pm 0.02$ |
| | FedAvg [TabNet] | - | - | - | - | - | - |
| | SplitNN [TabNet] | - | - | - | - | - | - |
| | FedTree | - | - | - | - | - | - |

Table 5: **All Classification results of FL experiments.** Here, we show the detailed performances of FC, FedAvg, and SplitNN in all three FL settings. It can be seen that FCs, while being much more flexible than our baselines, still achieve competitive or better results on various classification tasks.

| | CelebA | Imagenet32 | Imagenet |
|---|---|---|---|
| EiNet (Peharz et al., 2020a) | 5.37 | 5.74 | 6.28 |
| PyJuice (Liu et al., 2024) | 5.56 | 5.75 | 6.27 |
| FedPC (2 clients) | **5.31** | 5.57 | 6.24 |
| FedPC (4 clients) | 5.32 | 5.51 | 6.15 |
| FedPC (8 clients) | 5.35 | **5.49** | 6.13 |
| FedPC (16 clients) | 5.42 | 5.51 | **6.10** |

Table 6: **FedPCs outperform EiNets and PyJuice on density estimation tasks.** FedPCs achieve better results on density estimation tasks on three challenging image datasets (CelebA, Imagenet32 and Imagenet). This is because FedPCs can learn far larger models distributed across multiple machines. Results are reported as bits per dimension (bpd) averaged over 5 runs (lower is better). Note that we used Gaussian *densities* as PC leaves. Best value in **bold**, 2nd best underlined.

# E    COMMUNICATION EFFICIENCY

Communication efficiency is a critical property when it comes to learning models across multiple machines, as it is done in FL. Here, in addition to our theoretical results, we more intuitively provide further details on the communication efficiency of FCs. For that, we plot the communication cost in Megabytes (MB) required to train a FedPC vs. FedAvg/SplitNN in horizontal/vertical FL settings with datasets of different sizes (1M and 100M samples). Regardless of the number of samples in the dataset, FedPCs are more communication efficient compared to our baselines in both horizontal and vertical settings (see Fig. 5).

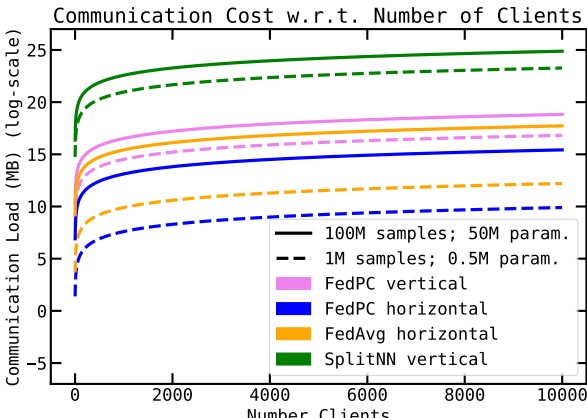

Figure 5: **FedPCs are communication-efficient.** We compare communication cost in Megabytes (MB) sent over the network during one full training of a model (0.5M/50M parameters) on a dataset (1M/100M samples) using results from Section 3.4. Results are shown on log-scale. It can be seen that FedPCs significantly reduce communication cost of training.

# F    EXPERIMENTAL DETAILS

## F.1    DATASETS

The following describes the datasets used in our experiments. If not stated differently, the datasets were distributed across clients as follows:

In horizontal cases, we either split samples randomly across clients (done for all binary classification tasks) or we distribute a subset of the dataset corresponding to a certain label (e.g. the 0 in MNIST) to one client.

In vertical cases, we split tabular datasets randomly along the feature-dimension, i.e. each client gets all samples but a random subset of features assigned. For image data, we split the images into non-overlapping patches which were then distributed to the clients.

In hybrid cases, we split tabular datasets along both, the feature and the sample-dimension. We do this s.t. at least two clients have at least one randomly chosen feature in commeon (but hold different samples thereof). For image data, we split images into overlapping patches, sample a subset of the dataset and assign the resulting subsets to clients.

**Income Dataset.** We used the Income dataset from `https://www.kaggle.com/datasets/ wenruliu/adult-income-dataset`. This dataset represents a binary classification problem with 14 features and approximate 450K samples in the train and 900 samples in the test set. We encoded discrete variables to numerical values using TargetEncoder from sklearn. Additionally, missing values were imputed using the median of the corresponding feature. Further we standardized all features.

**Breast Cancer Dataset.** We used the Breast Cancer dataset from `https://www.kaggle.com/datasets/uciml/breast-cancer-wisconsin-data`. It represents a binary classification problem with 31 features and 570 samples. We split the dataset into 450 training samples and 120 test samples. We standardized all features for training.

**Credit Dataset.** We used the Give Me Some Credit dataset from `https://www.kaggle.com/c/GiveMeSomeCredit`. The dataset represents a binary classification task with 10 features, 1.5M training samples and 100K test samples. We encoded discrete variables to numerical values using TargetEncoder from sklearn. Additionally, missing values were imputed using the median of the corresponding feature. Further we standardized all features.

**MNIST.** We used the MNIST dataset provided by pytorch. It contains 70K hand-written digits between 0 and 9 as 28x28 images (60K train, 10K test). We standardized all features as preprocessing.

**Imagenet/Imagenet32.** We used the Imagenet dataset provided by pytorch. It consists of about 1.2M images showing objects of 1000 classes. The images come in different resolutions; we resized each image to 112x112 (Imagenet) and 32x32 (Imagenet32) pixels, applied center cropping, and standardized all features as preprocessing. In our experiments, we used a pre-trained Vision Transformer (ViT) (Dosovitskiy et al., 2021) to obtain encodings of each image. Then, we applied KMeans to cluster the dataset into $n$ clusters (depending on the number of clients participating). Images of each cluster were then distributed to the clients, defining the client's datasets.

## F.2 HYPERPARAMETERS

The following tables show the setting of all relevant hyperparameters for each dataset and FL setting.

| FL-Setting | Dataset | Structure | Threshold | min_num_instances | glueing |
|---|---|---|---|---|---|
| horizontal | Income | learned | 0.3 | 200 | - |
| | Credit | learned | 0.5 | 200 | - |
| | Cancer | learned | 0.4 | 300 | - |
| vertical | Income | learned | 0.4 | 100 | combinatorial |
| | Credit | learned | 0.5 | 50 | combinatorial |
| | Cancer | learned | 0.4 | 300 | combinatorial |
| hybrid | Income | learned | 0.4 | 100 | combinatorial |
| | Credit | learned | 0.5 | 50 | combinatorial |
| | Cancer | learned | 0.4 | 300 | combinatorial |

Table 7: Hyperparameters used in our experiments for all tabular datasets.

| | MNIST | Imagenet(32) | CelebA |
|---|---|---|---|
| num_epochs | 5 | 25 | 10 |
| batch_size | 64 | 64 | 64 |
| online_em_frequency | 5 | 10 | 10 |
| online_em_stepsize | 0.1 | 0.25 | 0.25 |
| Structure | poon-domingos | poon-domingos | poon-domingos |
| pd_num_pieces | 4 | 4 | 4 |
| K | 10 | 120 | 120 |
| Leaf Distribution | Gaussian | Gaussian | Gaussian |
| min_var | $1 \cdot 10^{-3}$ | $1 \cdot 10^{-3}$ | $1 \cdot 10^{-3}$ |
| max_var | $1 \cdot 10^{-7}$ | $1 \cdot 10^{-7}$ | $1 \cdot 10^{-7}$ |

Table 8: Hyperparameters used in our experiments for image datasets.

## F.3 HARDWARE

All experiments were conducted on Nvidia DGX machines with Nvidia A100 (40GB) GPUs, AMD EPYC 7742 64-Core Processor and 2TiB of RAM.

