# OpenReview forum: "Scaling Probabilistic Circuits via Data Partitioning"
_ICLR.cc/2025/Conference — Submitted to ICLR 2025_

### Official Review · Reviewer_6dQr · 2024-11-02

**Soundness:** 2
**Presentation:** 1
**Contribution:** 2
**Rating:** 3
**Confidence:** 2

**Summary:**

This paper presents a novel method for scaling up probabilistic circuits (PC) by employing a divide-and-conquer framework. The approach involves training models locally on data subsets stored across multiple machines and subsequently aggregating these models through a tailored aggregation procedure. Unlike traditional parameterized neural networks, each model in PC represents a probabilistic distribution, necessitating the authors' development of a novel aggregation approach. Under certain assumptions, the proposed method can unify vertical and horizontal federated learning via a federated version of PC.

**Strengths:**

The approach of unifying federated learning (FL) through the lens of probabilistic circuits (PC) appears to be novel.

**Weaknesses:**

- **Clarity**: The paper is not clearly written, particularly in Section 3.3 (see detailed questions below).
- **Validity of Assumptions**: The framework relies on two main assumptions (Assumptions 1 and 2), but their practical relevance and applicability are unclear.

**Questions:**

- **Writing**
    - The definition of FC in Definition 3 is unclear. What is the specific difference between FC and PC based on this definition? In PC, the scope function corresponds to $\psi_{G}$, and the w function corresponds to either summation/product or density evaluation.
    - In Section 3.3, the communication network is not well-defined. How do the nodes N and their parent nodes correspond to different datasets? It’s therefore unclear how the proposed FedPC framework adapts to the FL framework.
    - Further clarification is needed in Section 3.3 on the following:
        - What does “all clients *share* their features with a server” mean?
        - What does “each random variable assumed to be *uniquely identifiable* across all clients” mean?
        - The notation used in Algorithm 1 is unconventional and confusing, for example, on Line 5.
        - What does it mean for “*the server to divide the joint feature space X into disjoint subspaces xxx using a unique descriptor vector u*”?
        - Overall, the description make it difficult to follow the proposed one-pass training approach.
- **Experiments**
    - In the experiment addressing Q1, it is unclear which density function is being estimated—is it the joint distribution of features and labels? Given the tractable inference property of PC, it might be more informative to leverage the estimated probabilities for conditional distribution P(y∣x) in order to perform classification on the image datasets. Without this, the motivation for applying PC to these tasks is less convincing.
    - The number of clients in the same experiment is quite small. How does the proposed method scale with an increasing number of clients?
    - In the experiment for Q3, is it appropriate to compare methods based on TabNet and FC? Could FedAvg show improved performance with an alternative architecture?
    - In the experiment for Q4, what is the rationale for considering only datasets split vertically and not horizontally?

---

> ### Author Response · Authors · 2024-11-18
> **Response to Reviewer 6dQr**
>
> We thank the reviewer for the thoughtful and detailed review. Also, we appreciate that the reviewer acknowledges our pursuit to unify horizontal, vertical and hybrid federated learning (FL) in one approach using federated circuits (FCs).
> We address the raised concerns below.
>
> All changes made in our manuscript during the rebuttal are marked in **green** (addressing reviewer 6dQr) and in **orange** (addressing all reviewers).
>
> > W1) Validity of Assumptions: The framework relies on two main assumptions (Assumptions 1 and 2), but their practical relevance and applicability are unclear.
>
> We discuss our assumptions and their necessity for our work in Section 3.1. Let us provide some more details on both assumptions here.
>
> Assumption 1 says that the distribution of a subset of random variables/features encountered in a distributed dataset is representable as a mixture distribution. This assumption is necessary since FedPCs (in Section 3.3), like PCs, model a distribution using mixture- and product distributions. Hence, Assumption 1 is frequently made in probabilistic modeling, although it is often not explicitly mentioned. If Assumption 1 does not hold, this implies that the data stored on the clients is not sufficient to learn the ground truth distribution.
>
> Assumption 2 states that it is possible to find clusters in the distributed dataset s.t. (some) random variables/features are independent. Similar to Assumption 1, this assumption stems from the fact that FedPCs, similar to PCs, represent distributions using mixtures and products. Akin to Assumption 1, Assumption 2 is frequently made in probabilistic modeling, although it is often not explicitly mentioned. Assumption 2 ensures that we can construct an FC structure capable of capturing dependencies among random variables residing on different clients.
>
> Thus, Assumptions 1 and 2 transfer standard assumptions from PCs to the case of distributed datasets.
>
> > What is the specific difference between FC and PC based on Definition 3? In PC, the scope function corresponds to phi, and the w function corresponds to either summation/product or density evaluation.
>
> Definition 3 is an extension of the standard PC definition. The semantic meaning of phi is kept in FCs (i.e., it refers to the scope function such as in PCs), and $\omega$ is a newly introduced function that maps each node N of an FC to a compute node. A compute node is simply a machine/client holding a dataset and participating in learning the FC.
>
> The key difference between PCs and FCs is thus that FCs are constructed over multiple machines, thereby defining a rooted DAG that not only determines the computation graph of calculating likelihoods but also determines a communication structure among clients during learning.
>
> We made this point clearer in our revision (see paragraph below Definition 3).
>
> > How do the nodes N and their parent nodes correspond to different datasets? It’s therefore unclear how the proposed FedPC framework adapts to the FL framework.
>
> Definition 4 in Section 3.3 aims to establish the connections between FC nodes and their semantic meaning. Note that only client nodes C correspond to a client holding a dataset and learning a local density estimator on the dataset held by a specific client. All other nodes are either sum or product nodes combining the local density estimators to a global density estimator over the entire distributed dataset. We added a more detailed description in our revision (see Definition 4).
>
> The connection between FedPCs (or FCs in general) and federated learning is established in Section 3.2 (Definition 1-3). Here, we show that horizontal, vertical, and hybrid FL can be reframed as a density estimation problem. The goal is to combine local density estimators learned over client datasets via the FC structure to form a global density estimator over the entire distributed dataset. Definitions 1 and 2 show that there is a direct correspondence between horizontal FL and learning mixture models as well as between vertical FL and finding independent clusters in a dataset s.t. densities over clusters can be combined by a product distribution. Hybrid FL then follows from a combination of Definition 1 and 2. As highlighted by Reviewer 4iCr, our reframing is a novel way to tackle FL problems through a unified approach.

---

> ### Author Response · Authors · 2024-11-18
> **Response to Reviewer 6dQr**
>
> > What does “all clients share their features with a server” mean? What does “each random variable assumed to be uniquely identifiable across all clients” mean?
>
> This means that each client shares a list of feature identifiers with the server. A feature identifier is, e.g., a unique name of a feature held by a client. For example, considering a finance dataset, a feature name could be “account_balance”.  As stated in line 257, we assume that each feature is uniquely identifiable across all clients, i.e., the same identifier/name corresponds to the same random variable in the data-generating process. This is necessary since the feature identifiers are used to construct the scopes of nodes in the FC. Clients do not share their data with the server, only identifiers corresponding to its features.
>
> To be more precise on that, we now say “all clients share their feature identifiers with a server” and introduce feature identifiers in our revision.
>
> > The notation used in Algorithm 1 is unconventional and confusing, for example, on Line 5.
>
> In line 5 of our pseudo-code, we follow standard set-theoretic notation. This line states that, for each client (row in $\mathbf{M}$), we collect the feature identifiers that exist on a specific client (all($\mathbf{u} == \mathbf{M}_{:, i}$)) where $\mathbf{M}$ is a binary indicator matrix where $\mathbf{M}_{i, j} = 1$ if client $i$ holds feature $j$. We added visual representation in our manuscript to illustrate the steps of Algorithm 1, thus making it more accessible (see Fig. 2).
>
> If anything else needs clarification, we kindly ask the reviewer to provide more details on which parts are perceived as unclear.
>
> > What does it mean for “the server to divide the joint feature space X into disjoint subspaces xxx using a unique descriptor vector u”?
>
> This step aims to group clients that share subsets of features. The elements of the binary vector $\mathbf{u}$ (which is a column from $\mathbf{M}$; see above) in line 5 in Algorithm 1 are 1 if a certain client $i$ holds a feature $j$. The server uses this information to first group together clients that share feature subspaces (corresponds to a mixture in the FedPC) and to group together clients holding distinct feature sets (corresponds to products of mixtures in the FedPC). Refer to Fig. 2 for a visual representation of Algorithm 1.
>
> > In the experiment addressing Q1, it is unclear which density function is being estimated—is it the joint distribution of features and labels?
>
> Following standard practices in density estimation, we estimated the joint distributions over features and labels in the case of tabular data (i.e., Income, Cancer, and Credit in Table 1), while we estimated the density over pixels in the case of image data (MNIST in Table 1) [Peharz2020, Dang2022, Liu2024].
>
> > Given the tractable inference property of PC, it might be more informative to leverage the estimated probabilities for conditional distribution P(y∣x) in order to perform classification on the image datasets. Without this, the motivation for applying PC to these tasks is less convincing.
>
> To stay consistent and ensure better comparability with other works considering density estimation, we followed standard practices and used log-likelihoods to compare FCs with our baselines in the image domain.
>
> Note, however, that we tested the classification capabilities of FedPCs in (Q3) and Figure 3, following standard practices from federated learning research. In the classification experiments we computed $p(y|x)$ and used it to predict the class of a data instance. We find that FedPCs achieve state-of-the-art classification results in all FL settings on a diverse and challenging set of tasks (low data regime [<1K samples], medium data regime [~300K samples], and large data regime [> 1M samples]).

---

> ### Author Response · Authors · 2024-11-18
> **Response to Reviewer 6dQr**
>
> > The number of clients in the same experiment is quite small. How does the proposed method scale with an increasing number of clients?
>
> Let us answer this question from a theoretical and a practical point of view.
>
> **Theory.** In Section 3.4, we analyze the communication cost of FedPCs. Our analysis reveals that FedPCs scale better than most existing horizontal and vertical FL approaches leveraging neural networks. For example, in horizontal FL, FedPCs are linearly more communication efficient than FedAvg-like algorithms since FedPCs do not require regular synchronization of client models during optimization.
>
> In vertical FL settings, FedPCs are typically more communication efficient than SplitNN-based approaches because FedPCs only require a structure construction phase and an aggregation phase, while SplitNN-based approaches require sending hidden representations of the data across the network in each optimization step.
>
> Appendix D visually compares the communication costs of FedPCs, FedAvg, and SplitNN.
>
> **Practice.** In practice, vertical and hybrid FL typically involve only a few clients because the dataset is split feature-wise (and additionally sample-wise in hybrid FL) across clients. Since datasets usually contain significantly more samples than features, it is natural that the number of clients in vertical and hybrid FL is relatively small.
>
> In horizontal FL, however, a larger number of clients typically participate in training. Since the leaves of FedPCs are learned independently and FedPCs are highly communication efficient in horizontal FL, only the computing power of the clients can be a bottleneck.
>
> On the modeling side, note that the primary goal of our work is to scale up PCs using FL. Since PCs have been trained on single machines so far, FedPCs have achieved a remarkable scaling of PCs.
>
> **References**
>
> [Peharz2020] Peharz et al. Einsum Networks: Fast and Scalable Learning of Tractable Probabilistic Circuits. ICML 2020.
>
> [Dang2022] Dang et al. Sparse Probabilistic Circuits via Pruning and Growing. NeurIPS 2022.
>
> [Liu2024] Liu et al. Scaling Tractable Probabilistic Circuits: A Systems Perspective. 2024.

---

> > ### Author Response · Authors · 2024-11-21
> > **Any further questions?**
> >
> > Dear Reviewer,
> >
> > We hope to have resolved all your concerns. If there are any further comments from your side, we will be happy to address them before the rebuttal period ends. If there are none, then we would appreciate it if you could reconsider your rating.
> >
> > Regards,
> >
> > Authors

---

> > > ### Comment · Reviewer_6dQr · 2024-11-25
> > >
> > > Thank you for your response.
> > >
> > > I still have concerns about Assumption 2. While I understand that this assumption was also used in PCs and that similar independence assumptions are common in variational inference (VI), VI often suffers degraded performance when the dependencies between random variables are strong. Given this, how does your method perform when the dependencies between random variables are high?
> > >
> > > Additionally, I may have missed the result you mentioned in your response: "In the classification experiments, we computed xxx. We find that FedPCs achieve state-of-the-art classification results in all FL settings xxx." If I understand correctly, your results are on tabular datasets. How does your method perform on vision tasks?

---

> ### Author Response · Authors · 2024-11-25
>
> Thank you for participating in the discussion!
>
> >Given this, how does your method perform when the dependencies between random variables are high?
>
> Assumption 2 states that it is possible to find **clusters** (i.e., subsets) in the (distributed) training data such that random variables within these clusters get independent. Thus, we do not assume that random variables residing on different clients are independent. We only demand that we can find clusters in which this independence holds.
>
> Assumption 2 motivates how the FedPC structure is built in Algorithm 1. Here, we first cluster the data on each client and hope to find clusters in which the random variables that are **not** shared among clients exhibit relatively high independence. These non-overlapping subsets of random variables are then represented as product nodes in the FedPCs structure, where we introduce **one product node per cluster**. The clusters are then aggregated in a sum-node (mixture), thus allowing FedPCs to capture dependencies in random variables even if the random variables are distributed across multiple clients.
>
> In practice, this approach works quite well. We analyzed the Income dataset for dependencies among features and found 6 features with relatively high correlations w.r.t to at least one other feature (Pearson correlation-coefficient $\rho$ was between 0.2 and 0.6). Note that the Income dataset has 13 features, thus approximately half of the features exhibit non-negligable correlations.
> In our experimental evaluation of FedPCs in vertical and hybrid FL settings, the set of correlating random variables was split into subsets. Each of the resulting subsets resided on a different client, thus the only chance FedPCs had to capture these dependencies was by leveraging the FedPC structure of mixtures of independent clusters as described above and in Sec. 4.
> Our empirical results on density estimation tasks (Tab. 1) and classification tasks (Fig. 4) in which Income was used as a large-scale dataset demonstrate that FedPCs effectively handle cases in which correlating random variables are distributed across multiple clients since FedPCs achieve state of the art results on density estimation and classification in hybrid and vertical FL scenarios.
>
> > How does your method perform on vision tasks?
>
> Additionally to our classification results on tabular data, we also evaluated FedPCs on an image classification task. Therefore, we perform classification on CelebA where we predict a binary label $\mathbf{y}$ given an image $\mathbf{x}$. We find that FedPCs achieve slightly better accuracy than Einets, demonstrating that FedPCs can also reliably solve classification tasks in the image domain (see table below).
>
> Note that while FedPCs achieve a relatively large gap to centralized models on density estimation tasks, the gap is not that big in classification tasks. We suspect that this is because density estimation is a harder task than classification: While in density estimation, one aims to learn the joint distribution over a large set of random variables as exact as possible (on CelebA we have 64x64x3 random variables corresponding to pixels), in classification it is enough to find a model that successfully discriminates between a small set of classes (2 in case of CelebA). Thus, in density estimation tasks, the upscaling effect of FedPCs is much more prevalent than in classification tasks.
>
> | Einet | FedPC (2 clients) |
> | ----- | ----------------- |
> | 0.87  | 0.89              |
>
> We hope to have cleared any concerns you have and hopefully you can reconsider your score.

---

### Official Review · Reviewer_MxNe · 2024-11-03

**Soundness:** 3
**Presentation:** 3
**Contribution:** 3
**Rating:** 6
**Confidence:** 1

**Summary:**

This paper introduces a novel distributed probabilistic circuit model named Federated Circuits (FC). The authors propose a data-partitioning approach that allows probabilistic circuits to scale effectively within distributed environments, improving both scalability and training efficiency. FC unifies horizontal, vertical, and hybrid federated learning setups and reframes federated learning as a density estimation problem. The experimental results demonstrate the superiority of FC across multiple large-scale datasets, showing its effectiveness in density estimation and classification tasks compared to existing methods.

**Strengths:**

- The proposed Federated Circuits (FC) offers a fresh perspective by redefining federated learning within a density estimation framework, enabling probabilistic circuits to scale in distributed learning environments.
- This method naturally handles horizontal, vertical, and hybrid federated learning setups, providing a flexible and efficient model expansion pathway.
- Experimental results indicate that FC outperforms or is comparable to existing neural network and tree-based methods

**Weaknesses:**

Since I am not very familiar with this field, I will refrain from commenting on the weaknesses. My confidence rate is 1, so please feel free to disregard my review.

**Questions:**

NA

---

> ### Author Response · Authors · 2024-11-18
> **Response to Reviewer MxNe**
>
> We thank the reviewer for the positive feedback highlighting the novel perspective of FCs on federated learning and our positive empirical results.
>
> If any question comes up during the discussion phase, we are happy to answer it.

---

### Official Review · Reviewer_4iCr · 2024-11-08

**Soundness:** 3
**Presentation:** 3
**Contribution:** 3
**Rating:** 8
**Confidence:** 3

**Summary:**

The paper proposes a PC-based approach for performing horizontal, vertical, and hybrid FL. To achieve this, each component of a partitioned data set is assigned to a leaf of a learned SPN. For those leaves sharing a common feature set, a sum node is introduced to aggregate the local distributions. Otherwise, a product node is utilized to combine distributions over disjointly featured random variables.

Overall, the paper is clearly written, and the proposed method (FedPCs) seems to be quite effective in both density estimation and classification tasks. Also, to the best of my knowledge, this is the first work simultaneously addressing horizontal, vertical, and hybrid FL in a single framework.

However, please keep in mind that I am mostly unfamiliar with the literature on probabilistic circuits. I look forward to engaging with the authors during the discussion period to potentially increase my score.

**Strengths:**

1. The proposed method is precisely described. Also, authors provide computer code for reproducing their experiments.

2. Assumptions are clearly stated, and the work’s novelty is well-established. Limitations are also partially addressed.

3. The demonstrated relationship between PCs and FL is insightful and may foster interesting future works in the field.

4. FedPCs lead to drastically faster learning while achieving comparable performance to a centralized approach.

5. A rough analysis of the method’s communication cost is provided.

**Weaknesses:**

1. Is Figure 1 correct? If I understood Section 3.3 correctly, a sum node is attached for each feature (subspace) shared by more than one client (lines 262-264). However, the equally featured partitions $\mathcal{P}_{1}$ and $\mathcal{P}_{2}$ are joined by a *product node* in Figure 1. I believe an illustration of Algorithm 1 would greatly improve the readability of Section 3.3.

2. It is trivial fact that conditional independence does not imply (marginal) independence, e.g., if $X_{i} = Y + \epsilon_{i}$ for some random variable $Y$ and white noise $\epsilon_{i}$, then $X_{1}$ and $X_{2}$ are conditionally (on $Y$) but not marginally independent. In this case, is Proposition 1 really necessary?

3. Table 2 suggests that increasing the number of clients concomitantly decreases the running time and enhances the performance of the learned model. Conventional wisdom, however, suggests that there should be a trade-off between these quantities. I thus wonder how many clients would be considered excessive. From a distributed learning perspective, a (possibly empirical) discussion on how to select the number of partitions  (e.g., with a validation set) would significantly strengthen the work.

4. Definitions 1 and 2, albeit standard, are somewhat cryptical. A concrete example of a distance metric $d$ would be helpful. Also, notations could be made clearer; although expressions such as $\mathbf{X}\_{c} \cap \mathbf{X}\_{c}$ and $\int_{\mathbf{X} \setminus \mathbf{X}\_{c}} p(x)$ are understandable, authors should decide whether $\mathbf{X}\_{c}$ is a set or a random variable.

5. A more extensive discussion on the method’s limitations would be appropriate. When does it fail? Given sufficient data, can FedPCs be scaled to an arbitrary number of clients? In other words, is there a computational bottleneck? Section 4 (Q2) indicates that the only limitation to arbitrarily scaling the model is statistical (via overfitting), rather than computational. On the other hand, is it a better option than traditional neural-based density estimation?

### Further comments

1. Proposition 2 in Appendix A.2 has no counterpart in the main text.

2. I am curious about how this approach compares to alternative neural-based density estimation methods, e.g., normalizing flows.

**Questions:**

See weaknesses above

---

> ### Author Response · Authors · 2024-11-18
> **Response to Reviewer 4iCr**
>
> We thank the reviewer for the insightful review. We appreciate that our work was perceived as novel and that the reviewer acknowledges the effectiveness of federated circuits (FCs) for upscaling probabilistic circuits (PCs) and federated learning (FL).
> Below, we address the questions raised in the review.
>
> All changes made in our manuscript during the rebuttal are marked in **blue** (addressing reviewer 4iCr) and in **orange** (addressing all reviewers).
>
> >  Is Figure 1 correct? If I understood Section 3.3 correctly, a sum node is attached for each feature (subspace) shared by more than one client (lines 262-264).
>
> Thank you for pointing this out! The observation is absolutely correct and we fixed Fig. 1 in our revision.
>
> > I believe an illustration of Algorithm 1 would greatly improve the readability of Section 3.3.
>
> We agree that an illustration of Algorithm 1 improves clarity. Thus, we added a figure (see Fig. 2) that visually depicts the steps of Algorithm 1.
>
> > In this case, is Proposition 1 really necessary?
>
> Although Proposition 1 is a known fact, we found it important to highlight it before introducing FCs and our one-pass training algorithm since the construction of FCs relies on the fact stated in Proposition 1. However, we renamed “Proposition 1” to “Fact 1” to indicate that it is no new theoretical result.
>
> > Table 2 suggests that increasing the number of clients concomitantly decreases the running time and enhances the performance of the learned model. Conventional wisdom, however, suggests that there should be a trade-off between these quantities. I thus wonder how many clients would be considered excessive. From a distributed learning perspective, a (possibly empirical) discussion on how to select the number of partitions (e.g., with a validation set) would significantly strengthen the work.
>
> We agree that there is a trade-off between number of partitions/clients (and thus training time) and model quality. How many clients are considered “excessive” is highly data- and application-dependent. For example, if the goal is mere upscaling of PCs on a dataset with millions of samples, the number of clients can be higher than for a dataset with only a few 10k samples. This is because each client needs a sufficiently large subset to perform density estimation reliably. One strategy for such use cases is to first cluster the dataset (possibly with a distributed clustering algorithm) such that each cluster holds approximately the same number of samples. Then, each cluster can be assigned to a client for density estimation. We found empirically that for complex datasets such as Imagenet 8-16 clusters (and thus 8-16 clients) provide a good datasplit, i.e., each client holds enough data to learn a high-performing density estimator while we achieve a significant speed up in training (see Table 2).
>
> On the other hand, for less complex datasets, a smaller number of clusters/clients (e.g., 2-4) provides a better trade-off since, for a higher number of clusters, each client does not hold a sufficiently large subset to learn a good density estimator. This effect can be seen in Table 2 (CelebA), where 2-4 clusters help improve the log-likelihood of the density estimate, while 8-16 clusters lead to worse density estimates. This is because CelebA is much less complex than Imagenet, hence, a higher number of clusters can lead to overfitting of the client’s density estimators.
>
> To demonstrate the upscaling capabilities of FCs, we **repeated our Imagenet and Imagenet32 experiments with 100 clients**. We find a **significant speed-up** in model training (25-30 times faster) while the **log-likelihood (LL) achieved by FedPCs was significantly better** than LL achieved by Einets and PyJuice (see table below). However, note that the LL achieved by FedPCs with 100 clients is worse than if 8 (Imagenet32) or 16 (Imagenet) clients were used. We attribute this to overfitting of some clients since the clustering step before training the FedPC yields a few clusters with a very small number of samples (20-100). We currently run experiments for CelebA with 100 clients as well and will add these results in the final version.
>
> |                     | Imagenet          | Imagenet32         |
> | ------------------- | ----------------- | ------------------ |
> | Einets              | -5893.59 $\pm$ 84 | 682.82 $\pm$ 3.5   |
> | PyJuice             | -5732.21 $\pm$ 71 | 664.54 $\pm$ 6.41  |
> | FedPC (16 clients)  | -1157.23 $\pm$ 74 | 1197.69 $\pm$ 10.8 |
> | FedPC (100 clients) | -1668.45 $\pm$ 78 | 1021.58 $\pm$ 9.84 |

---

> ### Author Response · Authors · 2024-11-18
> **Response to Reviewer 4iCr**
>
> > Definitions 1 and 2, albeit standard, are somewhat cryptical. [...] although expressions such as [...] and [...] are understandable, authors should decide whether X is a set or a random variable.
>
> Definitions 1 and 2 are a reframing of horizontal and vertical FL from a probabilistic point of view. We understand that from a FL point of view, the notation is untypical since our definitions consider the distribution space rather than the model parameter space when defining FL. However, we believe it is important to be exact in these definitions because the development of our one-pass algorithm and FCs relies on these definitions.
>
> Also, note that we now consistently denote random variables by $\mathbf{X}$ throughout the paper. To make notation clearer, we added a table in Appendix A providing an overview of all symbols used throughout the paper. Additionally, we added an intuitive explanation where appropriate in the paper.
>
> > A more extensive discussion on the method’s limitations would be appropriate. When does it fail? Given sufficient data, can FedPCs be scaled to an arbitrary number of clients? In other words, is there a computational bottleneck? Section 4 (Q2) indicates that the only limitation to arbitrarily scaling the model is statistical (via overfitting), rather than computational.
>
> Q2) and Table 2 in Section 4 indicate that, when using FCs to scale up PCs, one cannot use an arbitrarily large number of clients for training since each client requires a sufficiently large dataset to learn a high-performing density estimator. This is not surprising because any density estimator benefits from more datapoints, and splitting a fixed dataset across clients yields smaller subsets per client as the number of clients increases. If subsets get too small, this eventually leads to overfitting on the clients’ training data, hence deteriorating performance on the test set.
>
> However, in FL settings where each client brings its own dataset (i.e., the dataset grows with each participating client), the **only bottlenecks are the computational power of the clients and the communication network**. The more compute a client can use, the larger the leaf model learned by a client can be which can yield better density estimates. Regarding the communication network as a bottleneck, the higher the network bandwidth, the faster FedPCs can be trained. We have shown in Section 3.4 that **FedPCs are highly communication efficient** compared to alternative FL approaches (especially in horizontal FL), making them appropriate even in low bandwidth scenarios. Additionally, the aggregation of the FedPC into one model after training is no strict requirement because a FedPC model is naturally split across multiple machines, allowing for distributed inference. Thus, there is a trade-off between training cost and inference cost regarding communication efficiency.
>
>  > On the other hand, is it a better option than traditional neural-based density estimation?I am curious about how this approach compares to alternative neural-based density estimation methods, e.g., normalizing flows.
>
> This is an interesting question! FCs are a general framework, and in our experiments, we considered FCs to be parameterized with PCs. While PCs provide tractable inference, marginalization, and conditioning [Choi2020], neural-based density estimators have shown remarkable performance in inference (mostly) without providing tractable marginalization and conditioning. It is well known that neural-based density estimators like normalizing flows (NFs) are easier to adapt to high-dimensional and unstructured data such as images since NFs can be equipped with inductive biases such as convolutions to capture certain data properties better [Kingma2018]. This could be beneficial to obtain high-performing density estimators. So far, PCs lack these capabilities although work has been done to equip PCs with such inductive biases as well [Cory2019]. Since FCs are a general framework for density estimation across distributed datasets, they can benefit from both worlds: If the leaves of FCs are parameterized with NFs, one can learn NFs in a distributed fashion and benefit from the representational power of neural networks. If tractable marginalization is important, FC leaves can be parameterized by PCs (as shown in Section 3.3), and the resulting FedPC inherits all properties of the leaf PCs. Thus, the best parameterization of FCs depends on the application.
>
> **References**
>
> [Kingma2018] Kingma et al. Glow: Generative Flow with Invertible 1x1 Convolutions. NeurIPS 2018.
>
> [Cory2019] Cory et al. Deep Convolutional Sum-Product Networks. AAAI 2019.
>
> [Choi2020] YooJung Choi et al. Probabilistic Circuits: A Unifying Framework for Tractable Probabilistic Models. 2020.

---

> > ### Author Response · Authors · 2024-11-21
> > **Any further questions?**
> >
> > Dear Reviewer,
> >
> > We hope to have resolved all your concerns. If there are any further comments from your side, we will be happy to address them before the rebuttal period ends. If there are none, then we would appreciate it if you could reconsider your rating.
> >
> > Regards,
> >
> > Authors

---

> > > ### Comment · Reviewer_4iCr · 2024-11-25
> > >
> > > Thanks for your answer.
> > >
> > > I feel most of my concerns have been sufficiently addressed and will raise my score.
> > >
> > > On a side note, regarding weaknesses. In the embarrassingly parallel sampling literature, there are works pointing to a possibility of catastrophic failure when one of the models is poorly trained. Would this apply here?

---

> > > > ### Author Response · Authors · 2024-11-25
> > > >
> > > > Thank you for raising the score and participating in the discussion!
> > > >
> > > > Thank you for the hint! In our case, the quality of a model depends on various factors such as the data quality of the data residing on the clients, the amount of data, amount of clients, and the "weight" of each client (usually measured in the number of data samples). If we assume that all clients have approximately equally many high-quality samples and we train a FedPC on a moderate to high amount of clients, the overall FedPC model will still be of good quality because a poor model fit on one of the clients corresponds to a poor fit in only one of the model's leaves. In such a scenario, the other clients can likely counteract the poor fit of one leaf node. Hence, we believe that FedPCs are relatively robust in the failure cases of a few clients.

---

### Official Review · Reviewer_qUmr · 2024-11-10

**Soundness:** 3
**Presentation:** 2
**Contribution:** 2
**Rating:** 3
**Confidence:** 4

**Summary:**

The author propose a federated learning framework for probabilistic circuits learning. FCs unify horizontal, vertical, and hybrid federated learning by partitioning data and models across multiple machines, enabling efficient training and inference on large datasets.

**Strengths:**

1. The paper introduces federated learning framework to probabilistic circuits and use it for scaling up model training.

**Weaknesses:**

1. The evaluation in Table 2 is unclear. Since Gaussian distributions are used as PC leaves, it is not specified whether the reported values represent log-likelihoods or log-densities. If they are log-densities, they cannot be directly compared to benchmark log-likelihoods. To ensure a fair comparison, the values should be adjusted for discretization by converting log-densities into log-likelihoods.

2. The contribution is limited. Similar data partitioning methods were introduced in [1], and the idea presented in this paper is to assign different partitions to different machines/clients/nodes in order to scale up. However, the author does not clarify the distinction between the proposed method and that in [1].

3. The writing is redundant and notations are unnecessarily heavy. For instance, the federated circuits in Section 3.2 simply rephrase probabilistic circuits, but with the circuit nodes distributed across different clients.

[1] Robert Gens and Pedro Domingos. Learning the structure of sum-product networks. In Proceedings of the 30th International Conference on Machine Learning, 2013.

**Questions:**

See in [Weaknesses]
1. What is the inherent difference between your method and [1]
2. How do you evaluate performance in Table 2.

---

> ### Author Response · Authors · 2024-11-18
> **Response to Reviewer qUmr**
>
> We thank the reviewer for the feedback and appreciate that the scaling approach of our work was perceived positively.
> Let us address the raised concerns below.
>
> All changes made in our manuscript during the rebuttal are marked in **pink** (addressing reviewer qUmr) and in **orange** (addressing all reviewers).
>
> > W1, Q2) Since Gaussian distributions are used as PC leaves, it is not specified whether the reported values represent log-likelihoods or log-densities. If they are log-densities, they cannot be directly compared to benchmark log-likelihoods.
>
> To ensure a fair comparison, we parameterized all of the baselines and federated circuits (FCs) with Gaussian leaf distributions and thus reported log densities (see caption of Tab. 2). Hence, the comparison in Tab. 2 is valid. To improve clarity, we now additionally highlight this in Q2.
>
> > W2, Q1) Similar data partitioning methods were introduced in [1], and the idea presented in this paper is to assign different partitions to different machines/clients/nodes in order to scale up. However, the author does not clarify the distinction between the proposed method and that in [1].
>
> While the learnSPN algorithm presented in [1] learns the structure of Sum-Product Networks (SPNs) by partitioning data into smaller subsets, it is inherently different from our method of learning FCs. learnSPN **requires access to all data** samples and features and involves repeated clustering of the data (or subsets of the data) as well as repeated independence testing of feature sets. Thus, it is limited to relatively small-scale datasets and tabular data.
>
> In contrast, the one-pass training of FCs **does not require access to all samples and does not require repeated independence testing** when constructing the FC structure. This allows us to scale up PCs easily across machines and to align with the federated learning paradigm since no accumulation of a distributed dataset on a single machine is necessary (i.e., data is kept private). This would not be possible with learnSPN since clustering and independence testing require all samples and features to be accessible during learning. Also, note that our proposed one-pass training of **FCs supports arbitrary data modalities** since no independence testing is performed. Our one-pass training significantly contributes to scaling up PCs to real-world problems, and the experimental evaluation shows that the one-pass training of FCs achieves promising results on density estimation and classification tasks, often outperforming strong baselines for both families of tasks.
>
> > W3)  The writing is redundant and notations are unnecessarily heavy. For instance, the federated circuits in Section 3.2 simply rephrase probabilistic circuits, but with the circuit nodes distributed across different clients.
>
> We are aware that we introduce several definitions necessary for the development of FCs. However, as we are bridging two fields (federated learning and probabilistic circuits), we found it crucial to clearly and precisely introduce new concepts (such as FCs). This makes the paper self-enclosed and easier to understand for readers not familiar with PC or FL literature. We agree that Definition 3 is similar to the definition of probabilistic circuits. However, it is important to highlight the distributed nature of FCs with the scope function $\psi$ and the operation assignment function $\omega$ for a precise definition of FCs.
>
> **References**
>
> [1] Robert Gens and Pedro Domingos. Learning the structure of sum-product networks. In Proceedings of the 30th International Conference on Machine Learning, 2013.

---

> ### Author Response · Authors · 2024-11-21
> **Any further questions?**
>
> Dear Reviewer,
>
> We hope to have resolved all your concerns. If there are any further comments from your side, we will be happy to address them before the rebuttal period ends. If there are none, then we would appreciate it if you could reconsider your rating.
>
> Regards,
>
> Authors

---

> > ### Comment · Reviewer_qUmr · 2024-11-23
> >
> > Thank you for the response.
> >
> > My primary concerns persist. To the best of my knowledge, the standard practice for density estimation tasks is to report log-likelihoods rather than log-densities, regardless of whether the model is discrete or continuous.

---

> ### Author Response · Authors · 2024-11-23
> **Clarification**
>
> We thank the reviewer for participating in the discussion!
>
> We believe that there is a slight misunderstanding regarding log-likelihood and log density. Let us elaborate a bit further to clarify. For Tab. 2, we parameterized all models (i.e., FCs and the baseline models) with Gaussian leaves and optimized their parameters. The log-likelihood of a sample under a trained model is then $\ell(\theta; \mathbf{x}) = \text{log} p(\mathbf{x}; \theta)$ where $p$ is the model (i.e., an FC or one of the baseline models) and $\theta$ are the parameters of the model. All models used in the experiments for Tab. 2 were parameterized with Gaussian leaves, and their design ensures that they represent valid distributions; thus, $\ell$ **is the logarithm of a density value** of a valid distribution. We referred to it as the log density in our rebuttal above to highlight the fact that the model was parameterized with continuous random variables (Gaussian leaves).
>
> Note, however, that the log-likelihood is a function of the model parameters $\theta$ and **not** a function of the sample $\mathbf{x}$. Also, note that in our paper, in the caption of Tab. 2, we say that the values in Tab. 2 **"[...] are reported as log-likelihood values [...]"** which is consistent with existing literature (see[1] Tab. 1 and [2] Tab. 2).
>
> We hope that our explanation clarifies the reviewer's confusion.
>
> **References**
>
> [1] Peharz et al. Einsum Networks: Fast and Scalable Learning of Tractable Probabilistic Circuits. ICML 2020.
>
> [2] Yu et al. Characteristic Circuits. NeurIPS 2024.

---

> > ### Comment · Reviewer_qUmr · 2024-11-24
> >
> > I understand that the authors use Gaussian distributions as leaf nodes to parameterize the FC.
> >
> > However, since the FC models continuous distributions, the variable $l=\log p$ in your equation above is **the log of density value (log-PDF), not the log-likelihood**. Please report the log-likelihood values in bits-per-dimension of your trained models, at least on Imagenet32, so we can compare against existing approaches, for example Table 3 in [1].
> >
> > It is a standard practice to report log-likelihoods/bits-per-dimension for generative models representing continuous distributions [2].
> >
> > [1] Liu et al. Scaling tractable probabilistic circuits: A systems perspective. ICML 2024.
> >
> > [2] Kingma et al. Glow: Generative f low with invertible 1x1 convolutions. NeurIPS 2018.

---

> ### Author Response · Authors · 2024-11-24
> **Response to the reviewer**
>
> We want to thank the reviewer again for active participation in the discussion!
>
> The response helped to clarify the reviewer's concerns. We agree that reporting the log-likelihood in bits per dimension (bpd) improves the comparability of our work with other works. Following [1], we now provide the average bits per dimension in Appendix D (Table 6). Also, we included the results in our response in the table below. It can be seen that FedPCs outperform our baselines on all datasets (lower is better).
>
> Note that bpd does not exactly match the numbers reported in [1]. This is because our experimental setup is more challenging: We train all models on the full image (64x64 for ClebA, 32x32 for Imagenet32, 112x112 for Imagenet), while in [1], all models are trained on randomly chosen 16x16 patches.
>
> As said, we chose a different parameterization of the leave nodes (Gaussians instead of Binomials). This might also be a factor causing our results to differ slightly from the results in [1].
>
> |                    | CelebA | Imagenet32 | Imagenet |
> | ------------------ | ------ | ---------- | -------- |
> | Einet              | 5.37   | 5.74       | 6.28     |
> | PyJuice            | 5.56   | 5.75       | 6.27     |
> | FedPC (2 clients)  | 5.31   | 5.57       | 6.24     |
> | FedPC (4 clients)  | 5.32   | 5.51       | 6.15     |
> | FedPC (8 clients)  | 5.35   | 5.49       | 6.13     |
> | FedPC (16 clients) | 5.42   | 5.51       | 6.10     |
>
> We hope we have clarified your outstanding concern and that you can reconsider the score.
>
> **References**
>
> [1] Liu et al. Scaling tractable probabilistic circuits: A systems perspective. ICML 2024.

---

> ### Comment · Reviewer_qUmr · 2024-11-25
>
> 1. Why is there a significant gap between your reported results for Einet on the CelebA (5.37) and the numbers reported in the original Einet paper (−3.42 nats, equivalent to 4.93 bpd) [1]. Both implementations use Gaussian leaves.
>
> 2. Similarly, for Pyjuice, the gap on Imagenet32 between your reported result (5.75) and the original value (4.36) is around 1.4 bpd, it is too large to be explained by the patch model difference or leaf nodes distributions. [2]
>
> Overall, the baseline results reported here are too off away from original paper. Such inconsistency undermines the arguments in the paper. I recommend checking your evaluation or training pipeline. If the authors believe the evaluations are correct, then the results suggest that this method clearly does not demonstrate any empirical advantage.
>
> [1] Peharz et al. Einsum Networks: Fast and Scalable Learning of Tractable Probabilistic Circuits. ICML 2020.
>
> [2] Liu et al. Scaling tractable probabilistic circuits: A systems perspective. ICML 2024.

---

> > ### Author Response · Authors · 2024-11-25
> >
> > > Why is there a significant gap between your reported results for Einet on the CelebA (5.37) and the numbers reported in the original Einet paper (−3.42 nats, equivalent to 4.93 bpd) [1]. Both implementations use Gaussian leaves.
> >
> > We agree that both implementations use Gaussian leaves. However, our experimental setup is slightly different to better capture the federated learning aspect. In [1], the authors applied a pre-processing step in which CelebA was clustered into 100 clusters in the pixel space using K-Means. To ensure a  fair comparison between Einets and FedPCs, we set the number of clusters to 16 (the same as the highest number of clients/clusters in FedPCs). Thus, each cluster is larger and thus likely to be less "pure", ultimately leading to a slightly higher bpd score in our setup.
> >
> > >  Similarly, for Pyjuice, the gap on Imagenet32 between your reported result (5.75) and the original value (4.36) is around 1.4 bpd, it is too large to be explained by the patch model difference or leaf nodes distributions. [2]
> >
> > We aimed to settle on a unified and comparable experimental setup and thus adapted major parts of our setup from [1] (besides the mentioned differences). The different parameterization of the models and the patch model are the two major differences between our setup and the setup in [2]. Besides these differences, [2] also transform RGB values into YCoCg space which is done for better coding capabilities [3]. This transformation leads to results that "are not directly comparable to the results obtained from RGB images." [2]. Note that our primary aim is not to improve coding capabilities in the image domain. Rather, our aim is to introduce a new framework to learn probabilistic circuits in a federated fashion. Although the baseline bpd scores differ from the original papers, we still achieve a remarkable performance boost on large scale density estimation tasks compared to Einets and PyJuice.
> >
> > **References**
> >
> > [1] Peharz et al. Einsum Networks: Fast and Scalable Learning of Tractable Probabilistic Circuits. ICML 2020.
> >
> > [2] Liu et al. Scaling tractable probabilistic circuits: A systems perspective. ICML 2024.
> >
> > [3] H. Malvar and G. Sullivan. YCoCg-R: A Color Space with RGB Reversibility and Low Dynamic Range. 2003.

---

> ### Comment · Reviewer_qUmr · 2024-11-26
>
> I am not convinced by the authors.
>
> Maximizing the likelihoods of a continuous model with discrete RGB values without dequantization and using log-density as log-likelihood is fundamentally wrong. However, according to the authors' draft and response, this is what they have been doing. In Table 3, the authors maximize density at discrete points, resulting in positive log-densities. Proper dequantization should ensure log-density underestimates log-likelihood and cannot be positive. If dequantization is not performed, which is rare, the authors should integrate over the pixel bin to evaluate log-likelihood. Despite repeated discussion, the authors seem unaware of this critical distinction.
>
> In the revised version, the bits-per-dimension (bpd) evaluation is problematic. It deviates significantly from any existing literatures, including work on flow-based models [1] or probabilistic circuits [2]. On ImageNet32, bpd typically ranges between 3.9 and 4.4, yet the authors report values above 5.5 for the same baseline. Explaining it away as "slightly different experimental setups" or "data transformation" without further justification is unconvincing. This suggests the authors have not appropriately set or understood the baselines.
>
> Due to my concerns, I am keeping my score. The paper is not ready for submission. The authors did not properly train continuous models on discrete data, evaluate them correctly, or align their work with standard benchmarks and practices in the field.
>
> [1] Meng et al. ButterflyFlow: Building Invertible Layers with Butterfly Matrices. 2022.
>
> [2] Liu et al. Understanding the Distillation Process from Deep Generative Models to Tractable Probabilistic Circuits. 2023.

---

> ### Author Response · Authors · 2024-11-29
>
> When designing our experiments, we prioritized comparability over baseline performance. It is true that this leads to the performance in the experiments not matching the values reported in the original papers but we would like to stress that the setups are different. In that sense, our experiments do not show that we are achieving state-of-the-art performance on these datasets. On the other hand, our experiments do show that the federated approach does improve upon both EiNet and PyJuice under the given setup.
>
> We thus believe that the reviewer statement: "The paper is not ready for submission. The authors did not properly train continuous models on discrete data, evaluate them correctly, or align their work with standard benchmarks and practices in the field." is incorrect.

---

### Meta-Review · Area_Chair_dSak · 2024-12-21

**Metareview:**

The paper introduces Federated Circuits (FCs), a federated learning framework for probabilistic circuits learning.
FCs unify horizontal, vertical, and hybrid federated learning by partitioning both data and models across multiple machines.
Experimental results are provided to support the proposed method.

While the authors actively engaged with the reviewers during the rebuttal phase, they were unable to convince all reviewers—particularly Reviewers qUmr and 6dQr—of the significance of this work.
Concerns such as inappropriate experimental setups and significant limitations of the proposed approach remained unresolved.

As a result, I recommend rejecting the paper.

**Additional Comments On Reviewer Discussion:**

The reviewers raised the following concerns:

- Clarifications on some statements and/or experiments: Similar concerns were raised by all reviewers. However, I believe that only some of these issues were successfully addressed by the authors.
- Limitations of the proposed approach: These include the validity of the assumptions and the performance of the proposed approach beyond tabular data, as highlighted by Reviewers 4iCr and 6dQr.
While the authors made efforts during the rebuttal phase to address these concerns, they were not fully resolved.

I have considered all of the above points in making my final decision.

---

### Decision · Program_Chairs · 2025-01-22

Reject